# NF-κB oscillations translate into functionally related patterns of gene expression

Samuel Zambrano[1,2], Ilario De Toma[2], Arianna Piffer[2], Marco E Bianchi[1,2]*, Alessandra Agresti[1]*

[1]Division of Genetics and Cell Biology, San Raffaele Scientific Institute, Milan, Italy; [2]San Raffaele University, Milan, Italy

**Abstract** Several transcription factors (TFs) oscillate, periodically relocating between the cytoplasm and the nucleus. NF-κB, which plays key roles in inflammation and cancer, displays oscillations whose biological advantage remains unclear. Recent work indicated that NF-κB displays sustained oscillations that can be entrained, that is, reach a persistent synchronized state through small periodic perturbations. We show here that for our GFP-p65 knock-in cells NF-κB behaves as a damped oscillator able to synchronize to a variety of periodic external perturbations with no memory. We imposed synchronous dynamics to prove that transcription of NF-κB-controlled genes also oscillates, but mature transcript levels follow three distinct patterns. Two sets of transcripts accumulate fast or slowly, respectively. Another set, comprising chemokine and chemokine receptor mRNAs, oscillates and resets at each new stimulus, with no memory of the past. We propose that TF oscillatory dynamics is a means of segmenting time to provide renewing opportunity windows for decision.

**\*For correspondence:** bianchi. marco@hsr.it (MEB); agresti. alessandra@hsr.it (AA)

**Competing interests:** The authors declare that no competing interests exist.

## Introduction

Genetic circuits are instrumental for cells to provide adequate transcriptional responses to different external and internal stimuli, but we are still far from a complete understanding of how they work. The growing availability of single-cell measurements is bringing unprecedented insights into the dynamics of transcriptional responses: although it was traditionally thought that gene circuits should provide a temporally stable response to constant external stimuli, there is increasing evidence that the dynamics of gene circuits is more elaborate, and is often characterized by pulses of activity (*Levine et al., 2013*). A special case is oscillatory dynamics, in which the activity of some element in the genetic circuit varies periodically, and so does the output of the circuit. Paradigmatic oscillatory dynamics are associated with circadian clocks, which are present in a wide variety of organisms that synchronize their activities to the periodic variation in external daylight (*Bell-Pedersen et al., 2005*). However, oscillatory dynamics with periods far shorter than 24 hr have been reported in genetic circuits of bacteria (*Suel et al., 2007*), yeast (*Cai et al., 2008*; *Hao and O'Shea, 2012*) and mammalian cells (*Geva-Zatorsky et al., 2006*; *Hoffmann et al., 2002*; *Larson et al., 2013*; *Nelson et al., 2004*; *Shankaran et al., 2009*).

The dynamics of transcription factors in gene circuits can play a key role in information transmission through biochemical networks (*Selimkhanov et al., 2014*) by selectively modulating the expression of different genes (*Purvis and Lahav, 2013*). A paradigmatic example is how different p53 dynamics can selectively activate transcriptional programs that commit the cell to different cell fate decisions (*Purvis et al., 2012*).

**eLife digest** The process of producing useful biological molecules from genes – known as gene expression – is not always simple. Many genes are part of complex circuits, some of which show regular patterns of activity in response to an environmental cue. For example, the expression of some genes is tied to the 24-hour daily cycle of light and dark.

Transcription factors are proteins that control gene activation and expression, and some transcription factors periodically move in and out of the cell's nucleus – the compartment of an animal cell that houses the vast majority of the genetic material. This behavior is known as oscillation. A transcription factor called NF-κB oscillates, changing between an inactive form outside of the nucleus and an active form inside. NF-κB plays important roles in inflammation and cancer, and is activated by cues from outside the cell. Some of the genes that the active form of NF-κB activates then produce molecules that inactivate NF-κB, thus helping to establish the oscillations.

The benefits of the oscillations are not clear. However, recent studies suggest that environmental cues can cause small perturbations that gradually adjust the rate at which the oscillations occur, and in doing so, synchronize the oscillations amongst neighboring cells.

By using embryo cells from genetically engineered mice, Zambrano et al. investigated how NF-κB oscillations get synchronized. The experiments showed that the activity of the NF-κB protein and the expression of the genes it controls synchronize across neighboring cells whenever the external environmental perturbations come in pulses. However, once the pulsed cues stop, this synchronization is quickly lost. In essence, the cells reset after each environmental cue with no memory of previous episodes of NF-κB activity.

Further work revealed that the expression of the genes controlled by NF-κB also cycles and resets with each new environmental cue. However, the products of these genes accumulate in three different ways. Some accumulate quickly; some accumulate at a slow and steady pace; and some oscillate in amount, and this amount resets once the environmental cue has stopped. Each of these classes of gene products can be related to specific cell behaviors that activate sequentially on well-defined time schedules.

Overall, Zambrano et al. suggest that the ability of NF-κB to reset its activity with each new environmental cue gives cells the opportunity to pause and adjust course.

Zambrano et al. now plan to explore what happens to NF-κB synchronization in different cell types exposed to a collection of inflammatory stimuli. Along the same line, it will be worth exploring NF-κB behavior in cancer cells, where NF-κB activity is often out of control and drives unrestrained cell proliferation. These studies would contribute to a deeper understanding of cancer biology and to the identification of new treatments for the disease.

A property of free oscillating systems (or *free oscillators*, as analogy to simple mechanical oscillators such as the pendulum) is *entrainment*, by which an oscillator can gradually modulate its phase and frequency thanks to a small perturbation (*forcing*) that is itself periodic and oscillating with a period resonant with the intrinsic period of the oscillator (*Pikovsky et al., 2003*). A common forcing can also lead to the synchronous dynamics of multiple oscillators and thus to collective dynamical states. Entrainment was successfully reproduced in synthetic genetic oscillators, in which single bacterial cells (each expressing a fluorescent reporter protein in response to a synthetic genetic circuit) were entrained collectively to an oscillating provision of arabinose (*Mondragon-Palomino et al., 2011*). The forcing to one oscillator can also be provided by other oscillators, and this coupling can lead to the emergence of different collective dynamical states characterized by different synchronous dynamics (*Pikovsky et al., 2003*). Inter-cellular coupling has indeed been exploited to genetically engineer synchronous quorum sensing genetic oscillators in bacteria (*Danino et al., 2010*). On the other hand, intra-cellular coupling leads to locked oscillatory states for different cell oscillators, as recently shown for the circadian rhythm and the cell cycle (*Bieler et al., 2014*).

In mammalian cells, NF-κB is a typical transcription factor that displays intrinsic oscillatory behavior. NF-κB plays key roles in inflammation, immune responses, development and cancer (*Chaturvedi et al., 2011*; *Ghosh and Hayden, 2008*; *Karin, 2006*; *Ledoux and Perkins, 2014*;

*Naugler and Karin, 2008*). Strictly speaking, NF-κB is a family of dimers encoded by 5 different genes, but in what follows we will refer to p65/RELA independently of the dimer it forms. In resting cells, p65 exists mostly within a cytoplasmic complex bound to the IκB inhibitors (*Hoffmann et al., 2002*). Inflammatory signals like tumor necrosis factor alpha (TNF-α) or lipopolysaccharide (LPS) induce phosphorylation of IκB proteins by IKKs–upstream kinases in the signaling pathway–, ubiquitination and degradation of IκBs, and the release of active NF-κB that translocates into the nucleus to activate the expression of several genes, including those encoding for the IκB inhibitors (*Hoffmann et al., 2002*). Re-expression of IκBs contributes to relocate NF-κB in the cytoplasm, which is an inhibitory feedback loop for the system. A second feedback loop is centred on the protein A20 (*Ashall et al., 2009*), which upon activation of the system inhibits the IKKs. Thus, these two layers provide a nonredundant regulation of NF-κB response to external stimuli (*Werner et al., 2008*). Of note, the IκB inhibitors also show variable sensitivity to different stimuli and respond on different timescales to fine-tune the signaling pathway (*Paszek et al., 2010*; *Shih et al., 2009*). This system of negative feedback loops (summarized in *Figure 1A* upper panel) provides both a tight control of the response to external stimuli and flexibility. As a consequence of this complex wiring, upon constant stimulation the nuclear concentration of NF-κB in each cell oscillates with heterogeneous dynamics according to each cell's susceptibility and to the inherent stochasticity of the system (*Nelson et al., 2004*; *Tay et al., 2010*; *Zambrano et al., 2014a*). Due to such asynchrony at the single cell level, the NF-κB response appears almost non-oscillating at the cell population level (*Hoffmann et al., 2002*), and it is difficult to correlate NF-κB oscillatory profiles with gene expression outputs.

Obtaining synchronous NF-κB dynamics in cell populations is therefore important to study gene expression, and can give insights into the dynamics at a collective level. White's group showed that short trains of TNF-α pulses produce rounds of synchronous NF-κB translocation; the stimulation frequency affected both the translocation amplitude and gene expression levels (*Ashall et al., 2009*). A study performed while the present one was in progress indicated that entrainment of NF-κB oscillations at population level arises when 3T3 cells stably transfected with GFP-p65 are perturbed with regular trains of sawtooth-like profiles of TNF-α (*Kellogg and Tay, 2015*). Here, we used GFP-p65 mouse embryonic fibroblasts (MEFs) derived from knock-in mice and thus expressing physiological levels of p65 (*De Lorenzi et al., 2009*). Using a microfluidic device, we exposed cells to well-defined, periodic TNF-α stimuli, and eliminated continuously both catabolites produced by the cells and secreted proteins, which might generate a secondary autocrine/paracrine response. We find that GFP-p65 MEFs lock their NF-κB oscillations to the periodic external signal, and become synchronized. However, the 1:1 locking is maintained over a wide variety of frequencies of the driving TNF-α stimulus, does not improve over repeated stimulation and actually disappears fast when the external stimulus ceases, all of which suggest that entrainment is not achieved. The mathematical model we developed indeed suggests that NF-κB can behave as a damped oscillator, analogous to a mechanical damped harmonic oscillator; damped oscillators do not entrain but follow external forcing while it is present. Taking advantage of the fact that GFP-p65 MEFs under periodic stimulation behave as a synchronous population, we analyzed the transcriptional output at genome-wide level. We find that one single NF-κB dynamics translates into 3 different dynamics of transcriptional regulation, all of which can be reproduced by our mathematical model; the key discriminator is the parameter representing mRNA degradation. Furthermore, the three dynamical patterns correspond to specific functions, which suggests that group of genes were positively selected by evolution.

## Results

### Periodic forcing turns heterogeneous NF-κB oscillations into synchronous oscillations

To characterize the response of the NF-κB oscillatory system to different external stimuli, we made use of GFP-p65 knock-in MEFs (*De Lorenzi et al., 2009*; *Sung et al., 2009*; *Zambrano et al., 2014a*) cultured in a microfluidic device to control precisely the concentration and timing of the TNF-α stimulation (Materials and methods and *Figure 1—figure supplement 1A and B*). Notably, the flow rate in the microfluidic device is constant, so that the concentration of TNF-α cannot change due to the activity of the cells, nor can the medium accumulate catabolites or secreted proteins.

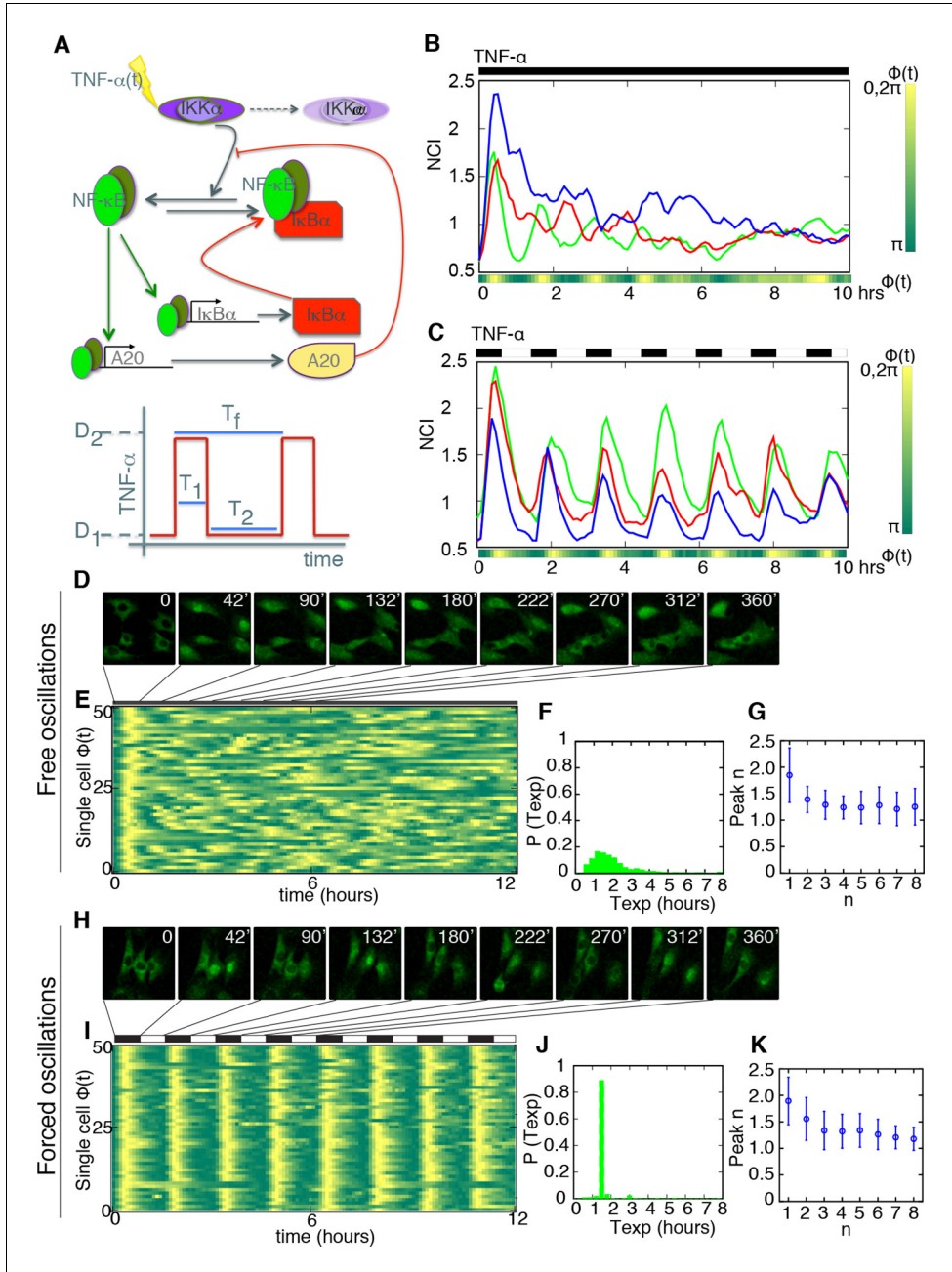

**Figure 1.** Periodic forcing turns damped heterogeneous oscillation into synchronous sustained oscillations. (**A**) The activity of NF-κB is regulated through different negative feedbacks provided by the inhibitors IκB and A20. The scheme at the bottom represents a generic forcing with periodically alternating TNF-α doses $D_1$ and $D_2$ of duration $T_2+T_1 = T_f$; $T_f$ is the period of the forcing. (**B, C**) Oscillations observed in three GFP-p65 cells obtained by computing the nuclear to the cytoplasmic GFP intensity (NCI) for constant flow of 10 ng/ml TNF-α (B) and upon alternating doses $D_1$=10 ng/ml TNF-α, $D_2$=0 ng/ml and $T_1$=$T_2$ =45 min (C). Each colour corresponds to a single cell trace. Oscillatory patterns can be effectively visualised using the phase ϕ of the oscillation, which is 2π in the maxima of the oscillatory peaks (yellow) and π in the local minima (green) in the colour phase-plot of ϕ(t) below the panels. Scale-bar for ϕ is on the right. (**D**) Time lapse images of cells under constant stimulation displaying the characteristic heterogeneous nuclear-to-cytoplasmic translocations. (**E**) Phase plot drawn for 50 cells, of 105 analysed, showing the asynchrony of the oscillations except for the first peak. (**F**) Distribution of the experimentally computed period of the oscillations $T_{exp}$, measured as the time between two consecutive oscillatory peaks. The distribution has a maximum at $T_0$ =90 min, which corresponds to the natural period. (**G**) Quantification of the height for each peak. (**H**) Time lapse images of the cells under periodic stimulation, showing synchronous NF-κB

*Figure 1 continued on next page*

*Figure 1 continued*

translocations between cytoplasm and nucleus. (I) Phase plot for 50 cells, of 206 analysed, showing a clear synchrony of the oscillations. (J) Distribution of the period of the oscillations $T_{exp}$. $T_{exp}$ corresponds almost perfectly to the period of the forcing. (K) Quantification of peaks height variation as described in G; values for n>1 are slightly higher than those observed under constant stimulation. Figure supplements from 1 to 10 are provided.

The following figure supplements are available for figure 1:

**Figure supplement 1.** Experimental set-up and quantification.

**Figure supplement 2.** Peaks and phase calculation.

**Figure supplement 3.** Damped oscillations for constant TNF-$\alpha$.

**Figure supplement 4.** Scheme of the simple ODE mathematical model used for NF-$\kappa$B dynamics, with the feedbacks provided by I$\kappa$B$\alpha$ and A20.

**Figure supplement 5.** Numerical exploration of the mathematical model suggests that damped oscillations are predominant.

**Figure supplement 6.** UV-photodamage is not detectable in imaged cells.

**Figure supplement 7.** Ongoing DNA repair is not detectable in cells imaged with Hoechst staining and UV irradiation.

**Figure supplement 8.** TNF-dependent activation of apoptosis in GFP-p65 cells stimulated with increasing doses of TNF-$\alpha$.

**Figure supplement 9.** UV and 488 laser imaging does not activate NF-$\kappa$B nor produce altered NF-$\kappa$B dynamics.

**Figure supplement 10.** GFP-p65 levels do not change in the cell population upon TNF-$\alpha$ stimulation.

**Figure supplement 11.** Cell cycle analysis of cells exposed to Hoechst and TNF-$\alpha$.

Constant stimulation of GFP-p65 knock-in cells in the microfluidic plate with a constant flow of 10 ng/ml TNF-$\alpha$ for up to 15 hr induced nuclear-to-cytoplasmic p65 oscillations (*Figure 1B*; *Video 1* and *Figure 1—figure supplement 2D*) that are qualitatively similar to the heterogeneous oscillations observed with static stimulation (*Sung et al., 2009*; *Zambrano et al., 2014a*).

Several controls were performed to exclude possible damaging effects related to imaging. Cells under constant flow of fresh medium without TNF-$\alpha$ divide and show almost no cell death (*Video 2*); cell death observed under continuous flow depends on the dose of TNF-$\alpha$ (*Figure 1—figure supplement 8* and *Video 1*) independently of the imaging conditions. UV-induced DNA damage due to imaging (*Cadet et al., 2005*) is negligible as assessed by immunostaining for thymine dimers (*Komatsu et al., 1997*; *Sinha and Hader, 2002*). DNA Damage Response (DDR) is also negligible as assessed by immunostaining of imaged cells for gammaH2AX (*Marti et al., 2006*; *Oh et al., 2011*; *Staszewski et al., 2008*) (*Figure 1—figure supplement 5* and *6*). Moreover, neither Hoechst exposure nor TNF-$\alpha$ affect dramatically the cell cycle (*Figure 1—figure supplement 11*). The controls to exclude possible effects of the nuclear dye (*Ge et al., 2013*; *Martin et al., 2005*) or photo-damage (*Cole, 2014*) are described in Materials and methods and in figure captions.

To analyse the oscillatory dynamics, we refined our recently published pipeline (*Zambrano et al., 2014a*) to compute the nuclear to cytoplasmic intensity (NCI) of GFP-p65 fluorescence for hundreds of cells per condition (Materials and methods and *Figure 1—figure supplement 1C*). Although this measure implies a partial segmentation of the cytoplasm and thus is more elaborate than the background-corrected mean nuclear intensity used recently by different authors (*Kellogg and Tay, 2015*; *Lee et al., 2014*; *Lee et al., 2009*; *Sung et al., 2014*) it is a ratio of intensities and thus it self-corrects for changes in image intensity in our experimental settings (see e.g. *Video 1* or *Video 2*).

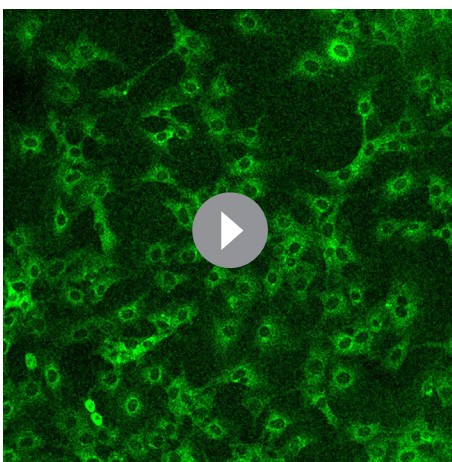

**Video 1.** Dynamics for constant flow of TNF-$\alpha$. Imaging of GFP-p65 knock-in cells in the microfluidic chamber stimulated with a constant flow of 10 ng/ml TNF-$\alpha$ for 12 hr. The stimulus induced nuclear-to-cytoplasmic p65 oscillations that are qualitatively similar to the heterogeneous oscillations observed with static stimulation. Along time, it is possible to appreciate the increase of TNF-induced cell death and apoptosis.

Furthermore, provided that the total amount of p65 is constant (*Figure 1—figure supplement 10*), NCI(t) is a monotonic function of nuclear NF-κB (*Figure 1—figure supplement 9B*) so that oscillations are observed in the former if and only if they are present in the latter (further details in Materials and methods). Finally, a rigorous description of the observed dynamics was achieved through the automatic identification of statistically significant peaks (Materials and methods and *Figure 1—figure supplement 2A and B*).

The timing between two consecutive peaks is the experimental period $T_{exp}$. $T_{exp}$ has a distribution with a maximum at 90 min (*Figure 1F*), which is the approximate intrinsic period reported for NF-κB oscillations that we called $T_0$ (*Nelson et al., 2004*; *Tay et al., 2010*; *Zambrano et al., 2014a*). As observed for static stimulation (*Zambrano et al., 2014a*), the observed oscillations are very heterogeneous (*Figure 1B*) and asynchronous (*Figure 1—figure supplement 3A* and *Figure 1—figure supplement 2D*). However, the variety of behaviors observed, including non-oscillating cells, is compatible with the dynamics observed in the same cells for static culture conditions, see (*Sung et al., 2009*; *Zambrano et al., 2014a*). Similar behaviors are found in the absence of nuclear staining (see *Figure 1—figure supplement 9C*), which led us to conclude that imaging conditions do not interfere with NF-κB signaling.

The tail in the period distribution (*Figure 1F*) and the observed average of four peaks in 12 hr of stimulation (*Figure 1—figure supplement 3B*), instead of the expected eight, indicate that in our cells the heterogeneous oscillations are damped and tend to converge to an equilibrium state under stimulation (*Figure 1B*). Indeed, although oscillatory peaks are observed for most of the cells, they are infrequent and irregular for times beyond 6 hr (*Figure 1B* and *Figure 1—figure supplement 2D*) as previously reported for the same cells in static conditions (*Sung et al., 2009*; *Zambrano et al., 2014a*). Hence, we describe here these heterogeneous oscillations as *damped* in contrast with the *sustained* oscillations, which continue regularly and unabated for a very long time in continuously stimulated cells, see for example (*Kellogg and Tay, 2015*).

Damped oscillations can easily emerge in the NF-κB genetic circuit. Indeed, a minimal deterministic model that takes into account the basic elements of the NF-κB genetic circuit (*Zambrano et al., 2014b*) (*Figure 1A*, *Figure 1—figure supplement 4*, see Materials and methods for a complete description of the model) shows that different combinations of the parameters can lead to different dynamics. The model parameters that we used for our explorations are provided in *Supplementary file 2*. We denote as $P_S$ those specifying the external signal and as $P_{NF-κB}$ those used to model the double IκB and A20 negative feedback; in our explorations, we allow them to vary differently depending on the associated uncertainty about their values (Materials and methods and *Supplementary file 2*). We generated a library of randomized parameters and found that the

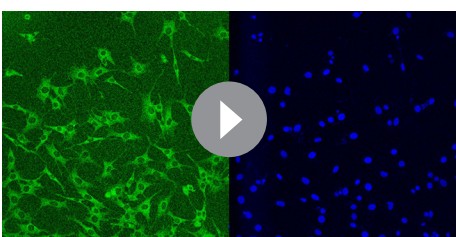

**Video 2.** Dynamics for Hoechst-stained cells in the absence of TNF-$\alpha$. Several controls were performed to exclude possible damaging effects related to imaging. This video shows that cells under constant flow of fresh medium without TNF-$\alpha$ divide and very few events of cell death are detectable. Left part: GFP channel; right part HOE channel. Twelve-hour imaging.

system presents a fixed point whose stability changes depending on the parameters (details on the stability analysis are found in the Materials and methods section). The vast majority of parameter combinations give rise to damped oscillations (when all the eigenvalues have all negative real parts, see *Figure 1—figure supplement 5A,B*). A smaller fraction of parameter combinations give rise to trajectories that converge to a stable limit cycle around the unstable fixed points (so certain eigenvalues have positive real parts, see *Figure 1—figure supplement 5A,B*). Interestingly, parameters for oscillating and non-oscillating cells are in similar intervals suggesting that it is the precise combination of the parameters, rather than a single one, what determines the resulting dynamics (see *Figure 1—figure supplement 5C*). Our simulations might also explain why other researchers found continuous periodic oscillations with $T_0 = 90$ min under a constant flow of TNF-$\alpha$ (*Kellogg and Tay, 2015*, and see Discussion) and the variety of damped oscillatory dynamics upon LPS recently reported for fibroblasts (*Cheng et al., 2015*) and macrophages (*Sung et al., 2014*). Our exploration shows further how variations of the parameters can give rise to a variety of dynamics that reflects what we find in an isogenic population (*Figure 1B* and *Figure 1—figure supplement 2*).

Considering the heterogeneity of dynamics, to better visualize the collective oscillatory state of the population in each condition, following *Mondragon-Palomino et al., 2011*, we computed and represented the phase of the oscillation $\phi(t)$ for each cell by detecting peaks and setting $\phi=0$ ($2\pi$) at the maximum of each peak and $\phi(t)=\pi$ in the minimum between two peaks (phase plots for the green time series are depicted in *Figures 1B,C*. See also Materials and methods and *Figure 1—figure supplement 2B,C*). Time series for single cells (*Figure 1B and C*) were converted to *phase plots* (*Figure 1E,I*) where each row represents one cell. Thus, oscillatory peaks can be easily observed. In the phase plot, the first response to constant TNF-$\alpha$ right after t=0 hr is synchronous in the population, but this synchrony is quickly lost, as previously reported (*Nelson et al., 2004*; *Tay et al., 2010*; *Zambrano et al., 2014a*).

To investigate the response of the NF-$\kappa$B oscillator to perturbations in the cell's environment, we switched periodically the stimulus concentration in the culture chambers in the microfluidics apparatus. TNF-$\alpha$ switching between doses $D_1$ and $D_2$ occurs in less than 1 min and generates a tightly controlled square profile of stimulation. Stimuli were applied for intervals of time $T_1$ and $T_2$. We refer to $T_f = T_1 + T_2$ as the *period of the forcing* and to $D_1 - D_2$ as the *amplitude of the forcing* (*Figure 1A*, lower panel).

We started our analysis by applying a periodic stimulation of 90 min, which is close to the intrinsic period of the NF-$\kappa$B oscillatory system. Single-cell traces are provided in *Figure 1C* for cells stimulated with $D_1 = 10$ ng/ml TNF-$\alpha$, $D_2 = 0$ ng/ml with $T_f = 90$ min. ($T_1 = 45$ min and $T_2 = 45$). Peaks are present in all the forcing cycles (*Figure 1C* and *Figure 1—figure supplement 3C*) and synchronous, in contrast to the asynchronous peaks observed in constantly stimulated cells (*Video 3*). Hence, we observe in this condition – a square forcing – a forcing-induced synchronous dynamics reminiscent of that obtained by applying short trains of pulses (*Ashall et al., 2009*). Our synchronous oscillations are visually apparent both in time-lapse images (*Figure 1H*) and phase plots (*Figure 1I*). $T_{exp}$ is sharply distributed around 90 min (*Figure 1J*), corresponding to the expected eight peaks in 12 hr (*Figure 1—figure supplement 3D*). The height of the first peak (*Figure 1K*) is similar to the height of the first peak under constant stimulation (*Figure 1G*), indicating that the experimental conditions are highly comparable. The height of the subsequent peaks is slightly higher for periodically stimulated cells but still heterogeneous.

Taken together, these results indicate that a periodic external stimulus can lock in step a population of cells in a long series of synchronous NF-$\kappa$B oscillations, in clear contrast with the remarkable oscillatory asynchrony and heterogeneity under constant stimulation.

## Oscillatory synchrony increases with the forcing amplitude

We then systematically analysed the response to variable forcing amplitudes, by applying different concentrations of TNF-$\alpha$. The phase plots obtained for TNF-$\alpha$ ranging from 10 to 0.1 ng/ml with $D_2 = 0$ ng/ml show that the coherence of the oscillations decreases with decreasing doses of TNF-$\alpha$ but still persists at low doses (*Figure 2A* and *Figure 2—figure supplement 1*). $T_{exp}$ is less sharply distributed around 90 min as the forcing amplitude decreases. Concomitantly, a second peak corresponding to multiples of $T_f$ becomes progressively more conspicuous (*Figure 2B*), suggesting that for lower doses of TNF-$\alpha$ an increasingly larger fraction of cells do not respond with a peak to some of the forcing cycles. In the plots of peak maxima (*Figure 2C*) the height of the first peak is not

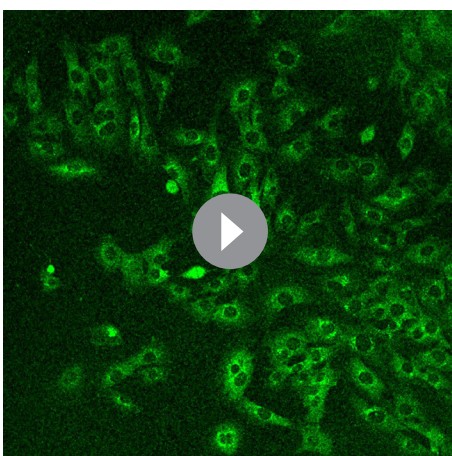

**Video 3.** Dynamics for $T_f$=90 min. Imaging of cells stimulated with $D_1$ =10 ng/ml TNF-$\alpha$, $D_2$ =0 ng/ml and $T_f$=90 min. ($T_1$=45 min and $T_2$ =45). Peaks are present in all the forcing cycles and synchronous. Twelve-hour imaging.

reproduced by the second and the third peaks, compatible with what reported for this forcing period (*Ashall et al., 2009*). However the heights converge under repeated forcing to a constant value for all the forcing amplitudes. This is typical of nonlinear dissipative oscillators in the presence of periodic forcing (*Goldstein et al., 2001*).

The degree of synchrony for each experimental setting of forcing can be evaluated considering the distribution of the *phase difference* $\Delta\phi$ between the timing of each oscillatory peak and the beginning of the forcing (see Materials and methods). For asynchronously oscillating cells under constant TNF-$\alpha$ stimulation, such distribution is flat (*Figure 2—figure supplement 2*). When cells are stimulated with different forcing amplitudes, the distribution of $\Delta\phi$ is narrower for higher doses of TNF-$\alpha$ (*Figure 2D*), indicating a higher degree of locking of the oscillations to the forcing for increasing amplitudes.

We then compared the degree of synchrony of the cell populations using the *synchrony intensity* $\eta$, an entropy-based quantifier of the distribution of the phase difference $\Delta\phi$ that is 0 for flat distributions and 1 for delta-like distributions (*Mondragon-Palomino et al., 2011*, and Materials and methods). Synchrony intensity increases with the dose, but is nonzero even when $D_1$=0.1 ng/ml TNF-$\alpha$ (*Figure 2E*). To understand if $D_2$=0 was a necessary condition for synchrony we applied $D_2$>0. Results show that the synchrony is maintained when $D_1$ is threefold $D_2$, while for smaller differences synchrony is almost lost (*Figure 2—figure supplement 3*).

We next investigated whether synchrony improved under repeated forcing. We computed $\eta_n$, the synchrony intensity at the *n*-th cycle of forcing (i.e. $\Delta\phi$ computed for peaks in the time intervals [(*n*–1) $T_f$, $nT_f$) for $n \geq$ 1). We find that $\eta_n$ does not increase (*Figure 2F*) in successive cycles of external forcing, meaning that the oscillations do not become more synchronous as the system is perturbed repeatedly.

Taken together, the above results indicate that NF-$\kappa$B dynamics adapts to varying amplitudes of periodic inputs. However the system does not seem to learn from the previous periodic forcing cycles.

## Cells oscillate synchronously following a variety of forcing amplitudes and periods

We then tested the adaptability of the NF-$\kappa$B system to different forcing periods $T_f$, ranging from $0.5T_0$ to $2T_0$.

We first considered periodic perturbation of $T_f$=$2T_0$=180 min, with $T_1$=30 min and $T_2$ =150 min, and TNF-$\alpha$ doses $D_1$ ranging from 10 ng/ml to 0.1 ng/ml, $D_2$ =0 ng/ml. Oscillations are locked to the forcing even for $D_1$ =0.1 ng/ml (*Figure 3A*, see also *Figure 3—figure supplement 1,A–C* and *Video 4*). The use of $T_1$=30 min assures the existence of sharp oscillatory and transcription peaks (see below) and also leads to synchrony of the oscillations for $T_f$=$T_0$ (see *Figure 3—figure supplement 3*, bottom panels). For all doses we find a bimodal distribution of the oscillation periods, with an overall maximum at 180 min and a much smaller relative maximum at 90 min (*Figure 3C*). This indicates that after a single pulse of forcing some cells oscillate a second time with a period similar to the intrinsic one. However, a closer analysis of peak maxima for time intervals of the form [(*n*–1)$T_0$, $nT_0$) = [(*n*–1)$T_f$/2, $nT_f$/2) (*Figure 3E*) reveals that peaks arising right after each periodic stimulation (even *n*) are higher than the rare ones (odd *n*) arising when stimulation is absent.

Oscillations in each condition show a good degree of synchrony, as described by the distributions of the phase differences $\Delta\phi$, which are remarkably narrow for all doses (*Figure 3G*); the synchrony intensity $\eta$ is nonzero even for $D_1$ = 0.1 ng/ml and increases with the dose (*Figure 3I*, blue line).

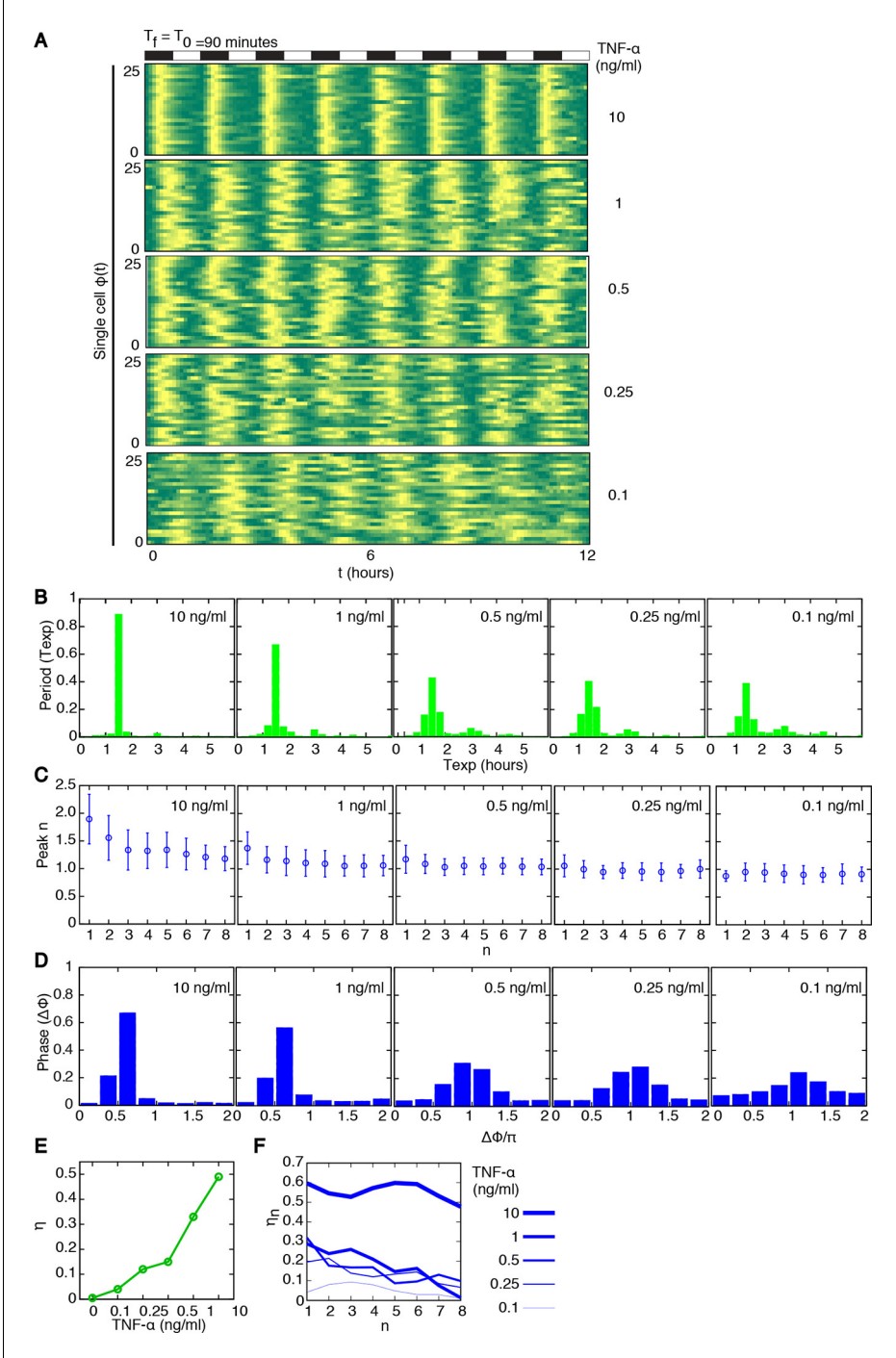

**Figure 2.** Synchronous oscillations arise for different forcing amplitudes. (**A**) Representative phase plots for 25 cells (out of 216, 80, 188, 263, 225 cells analysed) stimulated with $D_1$=10, 1, 0.5, 0.25, 0.1 ng/ml TNF-$\alpha$, $D_2$=0 ng/ml, for $T_f$ =90 min, $T_1$=$T_2$=45 min. (**B**) Distributions of the periods for the cells shown in panel (A); distributions become narrower as the dose $D_1$ increases. The appearance of a second peak at $T_{exp}$=3 hr at lower doses means that in some cycles a fraction of cells miss a peak and the interval to the next one is double. (**C**) Quantification of height of the n*th* peak in the different conditions considered in (A). By decreasing the stimulus amplitude, the ratio tends to stabilize to a constant value. (**D**) Distribution of the phase difference $\Delta\phi$ for the forcings considered: $\Delta\phi$ becomes narrower as the forcing amplitude is increased. (**E**) The synchrony intensity $\eta$, an entropy-based measure on how widely distributed the values of $\Delta\phi$ are, increases with the amplitude of $D_1$ for $T_f$ =90 min. (**F**) The
*Figure 2 continued on next page*

*Figure 2 continued*
synchrony intensity $\eta_n$ computed using only the peaks observed in each forcing cycle shows that the synchrony does not increase with the successive cycles of forcing. Figure supplements from 1 to 3 are provided.
The following figure supplements are available for figure 2:

**Figure supplement 1.** NCI and average dynamics for different forcings.
**Figure supplement 2.** Distribution of the phase differences for cells stimulated with constant TNF-$\alpha$.
**Figure supplement 3.** Dynamics of alternating doses $T_f$ = 90 min and $D_2 > 0$.

When $D_2 > 0$ ng/ml we found synchronized oscillations for $D_1$ at least three times $D_2$ (*Figure 3—figure supplement 2*).

We considered also the ability of the NF-κB system to respond to stimulations of periodicity below the intrinsic one. We selected $T_f = T_0/2 = 45$ min, but $T_1 = 30$ min led to a poor synchrony (see *Figure 3—figure supplement 3*, top panels). We therefore reduced $T_1$ to 22.5 min, and thus $T_2 = 22.5$, which proved to be a sufficiently long resetting of the external signal; in these conditions we obtained a sharply defined dynamical response. Oscillations are locked in step with $D_1 = 10$ and 1 ng/ml (*Figure 3B*, *Video 5*), whereas synchronization is weaker for $D_1 = 0.1$ ng/ml (this is also evident in the average NCI dynamics, see *Figure 3—figure supplement 1,D–F*). The experimentally determined oscillatory period $T_{exp}$ (*Figure 3D*) is narrowly distributed around $T_f = 45$ min for $D_1 = 10$ ng/ml, but we also find a small peak around $T_{exp} = 90$ min, indicating that some cells skip the oscillation elicited by the forcing. The peak height for each forcing cycle tends to stabilize to a constant value and to be lower for lower forcing amplitude, although the differences are small (*Figure 3F*). Also in this case, the synchrony correlates with the dose: in fact, the phase difference $\Delta\phi$ tends to be more narrowly distributed for higher values of $D_1$ (*Figure 3H*), and the synchrony intensity $\eta$ grows with the dose (*Figure 3I*, red line).

It has been recently reported that NF-κB oscillations can be synchronized by entrainment (*Kellogg and Tay, 2015*). However, in our experiments the synchrony intensity for each cycle, $\eta_n$, does not increase when the system is forced repeatedly, either with periods of 90, 180 or 45 min (*Figures 2F*, *3J,K*), in contrast to what is observed for entrained oscillators (*Mondragon-Palomino et al., 2011*). Furthermore, when oscillators are entrained by external forcing, their oscillations also show *m:n* resonant patterns (*m* oscillations for each *n* cycles of the forcing) in so-called Arnold tongues (*Pikovsky et al., 2003*). However, we only observed 1:1 synchronization patterns for all the periods of forcing (*Figure 3A,B*), with no clear evidence of the 2:1 and 1:2 patterns expected for entrainment. We thus investigated further whether the synchronization mechanism we observed does correspond to entrainment.

## The synchronization mechanism of GFP-p65 knock-in MEFs is not entrainment

Once the forcing ceases, entrained oscillators dephase gradually, losing their sharply defined common entrained oscillatory period. We then investigated whether our cells kept a memory of the synchronous dynamics once the periodic forcing ceased. The phase plots for 75 cells forced with $T_f = 45$ min $D_1 = 10$ ng/ml (*Figure 4A*, washing out after 16 forcing cycles), and for $T_f = 90$ min with $D_1 = 1$ ng/ml (*Figure 4B*, washing out after eight forcing cycles), indicate that when the external stimulus stops, cells lose quickly their synchrony. This is also evident from single-cell NCI traces and from the peaks detected in the same conditions (*Figure 4C and D* respectively). Indeed, only for $T_f = 45$ min we observed small peaks after the last cycle of the forcing, but these peaks are on average 90 min away from the last forced peak, and correspond to the intrinsic oscillatory period of our oscillator. The quick loss of synchrony can also be observed in the evolution of the synchrony quantifier $\eta_n$ for the two last cycles of the forcing (*Figure 4E*, n=1,2) and the two subsequent (virtual) ones (*Figure 4E*, n=3,4). This is also the case when considering cells forced with $T_f = 90$ min and a high stimulus amplitude $D_1 = 10$ ng/ml, after which we apply a constant flow of 10 ng/ml TNF-$\alpha$: cells present a last well-defined translocation (*Figure 4F and 4G*) and rapidly dephase (*Figure 4H*).

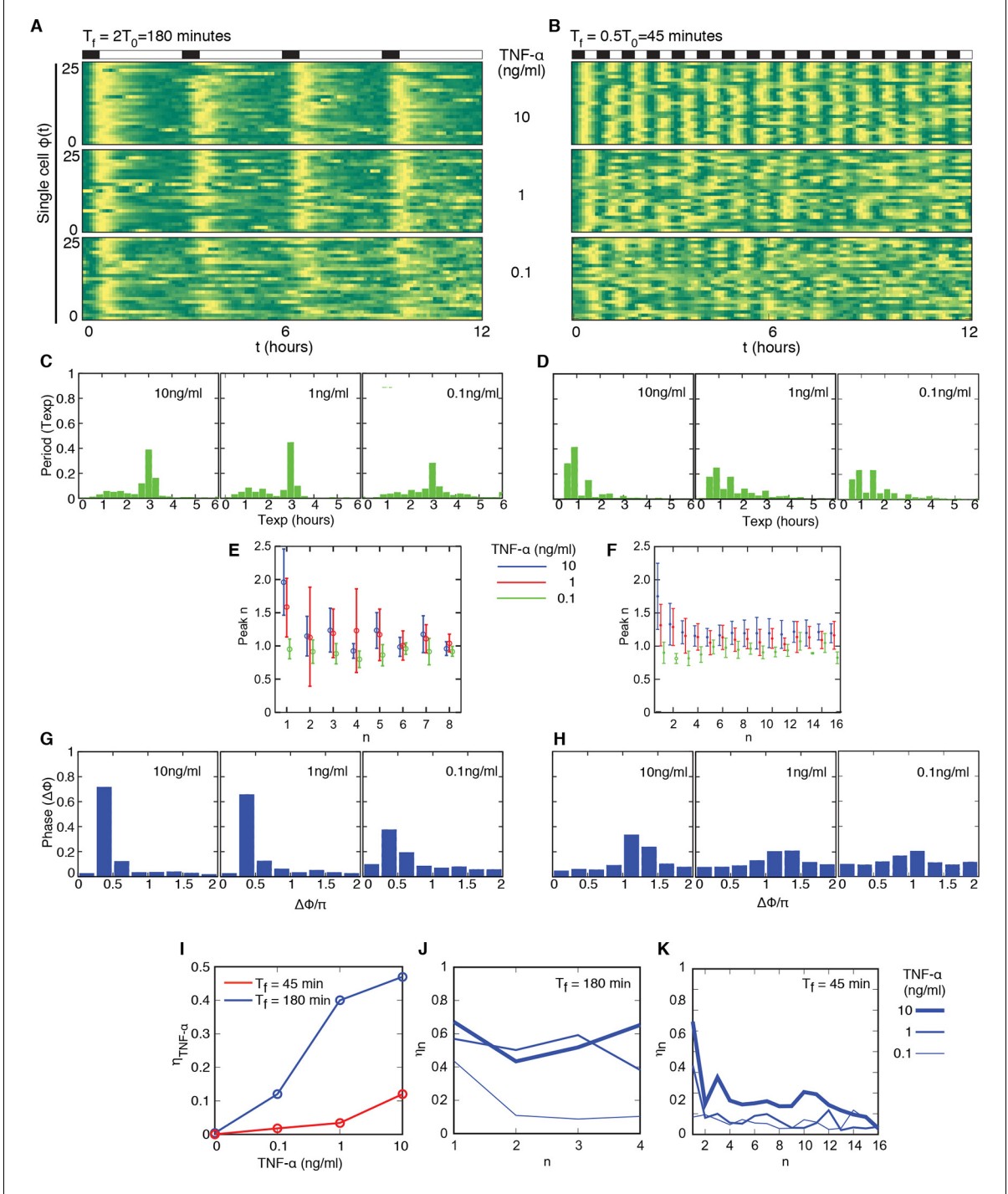

**Figure 3.** Cells adjust oscillations to different periods for a wide range of forcing amplitudes. Representative phase plots for 25 cells stimulated with $D_1=10, 1, 0.1$ ng/ml TNF-$\alpha$ (out of 151, 77, 123 cells analysed, respectively) $D_2=0$ ng/ml for (**A**) $T_f=180$ min with $T_1=30$ min and (**B**) $T_f=45$ min with $T_1=22.5$ min (analysed 101, 112, 119 cells). (**C, D**) Plots showing the distributions of the periods for the conditions given in panels (**A**) and (**B**), respectively. (**E, F**) Average peaks height in the intervals $[(n-1) T_0, T_0)$ and $[(n-1)T_0/2, nT_0/2)$, for A and B, respectively. In (**E**), even n correspond to peaks right after stimulation, odd n correspond to the small peak arising between two consecutive stimulations. (**G, H**) Distribution of the phase difference $\Delta\phi$ for the forcings in A and B: $\Delta\phi$ has narrower distributions for higher doses. (**I**) The synchrony intensity $\eta$ grows with the doses for $T_f=180$ min (blue) and $T_f=45$ min (red). (**J, K**) Synchrony intensity plots show that $\eta_n$ does not increase as successive cycles of forcing are applied to the system, both for $T_f=180$ min and $T_f=45$ min, respectively. All the analyses included all the tracked cells with no preselection of the responding ones. Figure supplements 1 to 3 are provided.

*Figure 3 continued on next page*

*Figure 3 continued*

The following figure supplements are available for figure 3:

**Figure supplement 1.** NCI and average dynamics for different forcings.

**Figure supplement 2.** Dynamics of alternating doses $T_f$=180 min and $D_2$>0.

**Figure supplement 3.** Dynamics for cells synchronised with $T_1$=30 min and different TNF concentrations.

*Kellogg and Tay (2015)* found no synchrony for $T_f$=60 min. However, we found that our cells can actually be synchronized under periodic stimulations with $T_f$=60 min ($T_1$ =$T_2$ =30 min) when $D_1$ =10 ng/ml (*Video 6*) and $D_2$ =0 ng/ml (*Figure 4—figure supplement 1A*). Furthermore, we found that synchrony is still visible for $D_2$>0 ng/ml (*Figure 4—figure supplement 1B–D*), as we had observed for other stimulation periods. The difference might arise from the fact that we use a forcing different from the sawtooth-like TNF-$\alpha$ profile used by *Kellogg and Tay (2015)*, which is obtained by periodically and quickly replacing the medium in contact with the cells with fresh TNF-$\alpha$. The sawtooth concentration profile is assumed to arise due to a clearance process of TNF-$\alpha$ by degradation and internalization. We then applied sawtooth forcing to our cells (*Figure 4—figure supplement 4*). Cells lock their oscillations to the sawtooth forcing of periods $T_f$=60 min and $T_f$=90 min at the doses considered. Interestingly, the synchronization is lost for $T_f$=180, suggesting that autocrine-paracrine signalling might introduce a distortion in the cell environment, which is reduced when the medium is changed more frequently. Alternatively, the different geometry of our microfluidic with respect to the one used in *Kellogg and Tay (2015)* might lead to a slower TNF-$\alpha$ degradation and produce profiles closer to static concentration than to oscillatory stimulation. The profiles observed are indeed similar to some of the dynamics observed for alternating doses of TNF-$\alpha$ with $D_2$>0 ng/ml, such as those for $T_f$=60 min (*Figure 4—figure supplement 1B–D*), $T_f$=90 min (*Figure 2—figure supplement 3*) and $T_f$=180 min (*Figure 3—figure supplement 2*), with compatible values of $\eta$. However, it is clear that with sawtooth forcing of $T_f$=60 min the synchrony is strong, at variance with the results reported by *Kellogg and Tay (2015)*.

Overall, we conclude that the synchronization mechanism that we observe is not entrainment. Rather, it is similar to that of simple mechanical damped oscillators, such as the damped harmonic oscillator, which after a perturbation tend to relax to an equilibrium state. When these mechanical systems are challenged with an external periodic forcing, the period of the oscillations tends to match the period of the forcing (*Pikovsky et al., 2003*). In fact, our minimal mathematical model of the NF-$\kappa$B circuit (*Figure 1—figure supplement 4*) reproduces damped oscillations (e.g. *Figure 1—figure supplement 5A,B,D*). In these conditions, the period of the periodically forced system converges to the period of the forcing (*Figure 4—figure supplement 2*). Of note, the model reproduces the small peaks appearing between two forcing cycles (*Figure 4—figure supplement 3*), similar to the small peaks that we observed for the same forcing as in *Figure 3A*.

When behaving as a damped oscillator, the NF-$\kappa$B system is well suited to quickly synchronize its oscillatory period to input signals of a wide variety of timescales. This is probably a desired feature for a system that underlies responses to environmental challenges, which do not come with a particular periodicity. This is the reverse of what might be desirable for circadian clocks, whose design should privilege the entrainment to the 24-hr light/darkness period. To understand how the oscillatory pattern of NF-$\kappa$B affects its

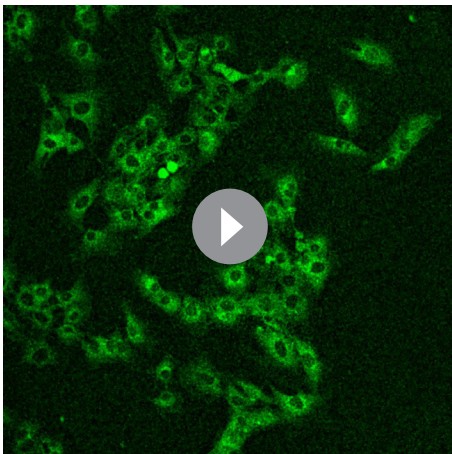

**Video 4.** Dynamics for $T_f$=180 min. Imaging of cells stimulated with $D_1$ =10 ng/ml TNF-$\alpha$, $D_2$ =0 ng/ml and $T_f$=$2T_0$=180 min ($T_1$=30 min and $T_2$ =150). Oscillations are locked to the forcing. Twelve-hour imaging.

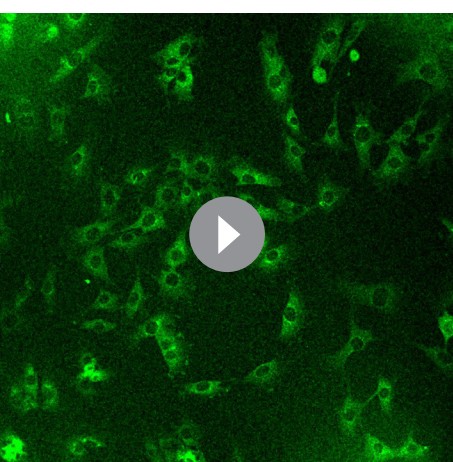

**Video 5.** Dynamics for $T_f$=45 min. Imaging of cells stimulated with $D_1$ =10 ng/ml TNF-$\alpha$, $D_2$ =0 ng/ml and $T_f$=$T_0$/2 =45 min ($T_1$=22.5 min and $T_2$ = 22.5 min). Such short wash-out provides a sufficient resetting of the external signal; in these conditions we obtained a sharply defined dynamical response and oscillations are locked in step. Twelve-hour imaging.

biological output, we focused on the transcription dynamics of genes controlled by NF-κB.

## Synchronous NF-κB dynamics reveal different patterns of gene expression

The heterogeneity of NF-κB dynamics in the cell population under constant stimulation makes it difficult to establish a direct correspondence between NF-κB activity and transcription. However, under conditions of synchronous oscillation the average dynamics corresponds to the dynamics of single cells. This is for example the case for cells stimulated with $D_1$ =10 ng/ml $D_2$=0 ng/ml TNF-$\alpha$, $T_1$=30 min and $T_2$ =150 min ($T_f$= $2T_0$) (*Figure 5A*). Numerical simulations with a simple mathematical model of transcription under the control of the regulatory network using parameters from our previous work (*Figure 5—figure supplement 1*, Materials and methods and *Zambrano et al., 2014b*) suggested that for some genes there should be a clear coordination between the input pulsed signal, the oscillating dynamics of NF-κB and the transcriptional output (*Figure 5B*).

To validate the model's prediction, we tested RNA transcription in the synchronous oscillatory condition shown in *Figure 5A*. RNA samples were prepared at time 0 and 20, 40 and 60 min after each TNF-$\alpha$ pulse. Quantitative RT-PCR was performed for both the nascent unspliced and the mature mRNA forms of the prototypical early and late genes IkBa and Ccl5/RANTES, respectively (*Figure 5—figure supplement 2* and Materials and methods). Nascent transcription of both genes starts synchronously with each TNF-$\alpha$ pulse and p65 nuclear localization (*Figure 5C,D*), and the levels of nascent transcripts clearly oscillate. Mature transcripts of the early gene IκBα follow the oscillatory dynamics, show transcription-coordinated splicing and do not accumulate after stimulus wash-out (*Figure 5C*). On the contrary, the mature form of the late gene Ccl5 accumulates slowly and progressively along 12 hr (*Figure 5D*).

This latter observation extends the notion that the splicing rate for inflammatory genes plays a key role in regulating the timing of gene expression, as previously reported (*Hao and Baltimore, 2013*). The induction levels for IκBα in the first hours hardly differ from those under chronic stimulation ((*Hao and Baltimore, 2013*); *Figure 5* and *Figure 5—figure supplement 2*). Interestingly, we also find oscillatory dynamics of IκBα and Ccl5 nascent transcripts when cells are stimulated with a forcing period of $T_f$=90 min, although the oscillations are less prominent than for $T_f$=180 min. The expression of Ccl5 for $T_f$= 90 min grows monotonically (*Figure 7—figure supplement 1*).

We then asked whether our minimal mathematical model (*Figure 5—figure supplement 1* and *Figure 1—figure supplement 4*) could simultaneously fit NCI dynamics (*Figure 5A*) and transcription profiles. Our model of transcription can be interpreted as a population-level NF-κB–driven telegraph model (*Suter et al., 2011*) that includes non-cooperative transcription activation by NF-κB (*Giorgetti et al., 2010*) and the role as transcriptional repressor reported for IκBα (*Arenzana-Seisdedos et al., 1995*; *Kellogg and Tay, 2015*; *Lipniacki et al., 2004*; *Lipniacki et al., 2006a*; *Lipniacki et al., 2006b*; *Tay et al., 2010*); we denote the parameters used to fit the transcription as $P_G$ (details in the Materials and methods section). The fittings were performed by keeping constant the parameters of the external signal $P_S$ (which is externally imposed) and using the same parameters regulating NF-κB dynamics, $P_{NF-κB}$ in *Figure 5A,E,F*, while using different individual transcription parameters $P_G$ for each gene (*Figure 5E,F*). Despite being much simpler than other existing models of NF-κB dynamics (*Ashall et al., 2009*; *Tay et al., 2010*), our model fits faithfully the observed transcription dynamics for both IκBα (*Figure 5E*) and Ccl5 (*Figure 5F*).

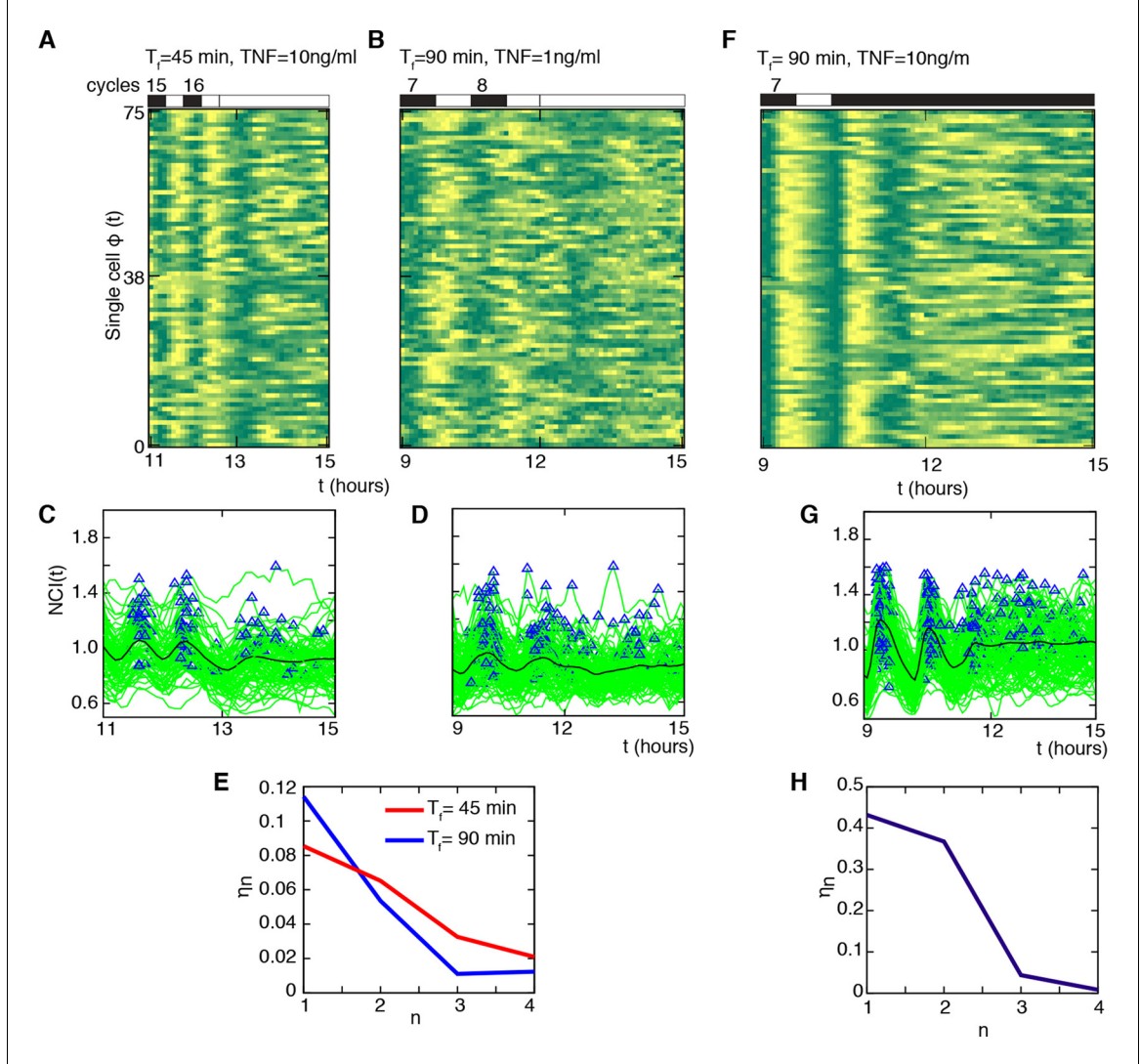

**Figure 4.** Cells do not keep a memory of the synchronous oscillatory dynamics. (**A, B**) Phase plots of the last two oscillation cycles for $T_f$ =45 min ($D_1$=10 ng/ml) and $T_f$ =90 min forcing ($D_1$= 1 ng/ml) (number of forcing cycles are indicated above) followed by a period of 3 hr with no stimulation (75 cells are displayed out of 106 and 197 cells analysed, respectively). (**C, D**) NCI time series at single cell level (green lines) for the two conditions (A) and (B). Blue triangles indicate the peaks considered in the computing. The thick black line is the average NCI, showing a small peak 90 min after the last forced peak for $T_f$ =45 min (C), compatible with the natural timescale of the free oscillations. (**E**) The synchrony intensity $\eta_n$ for the last two forced peaks (n=1, 2) and for the peaks detected in the absence of the stimulus (n=3, 4) illustrates the fast loss of synchrony. (**F**) Phase plots of the last two oscillation cycles for $T_f$ =90 min (number of forcing cycles are indicated above) ($D_1$=10 ng/ml) followed by 4.5 hr flow of 10 ng/ml TNF-$\alpha$ (75 cells are displayed out of 149 cells analysed). (**G**) NCI time series at single-cell level (green lines). Blue triangles indicate the peaks considered in the computing. The thick black line is the average NCI, showing a small peak 90 min after the last forced peak, compatible with the natural timescale of the free oscillations. (**H**) The synchrony intensity $\eta_n$ for the last two forced peaks (n=1, 2) and for the peaks detected in the presence of the stimulus (n=3,4) illustrates the fast loss of synchrony. Figure supplements from 1 to 4 are provided.

The following figure supplements are available for figure 4:

**Figure supplement 1.** Dynamics of alternating doses $T_f$=60 min and $D_2$>0.

**Figure supplement 2.** The model predicts that NF-$\kappa$B oscillations follow the forcing period.

**Figure supplement 3.** Interpretation of the $T_{numerical}/T_f$ ratios close to 0.5 in *Figure 4—figure supplement 2*.

**Figure supplement 4.** Sawtooth-like profiles lead to heterogeneous but synchronous dynamics in GFP-p65 cells.

Overall, our results show that the expression of two different genes can follow different dynamical patterns even if the entire cell population is effectively locked to the dynamics of the transcription factor that controls both genes.

## Transcription dynamics discriminates between groups of functionally related genes

Genome-wide profiling represents the logical further step to understand whether additional NF-κB regulated genes follow the diversified dynamics observed for Nfkbia/IκBα and Ccl5. We performed microarray analysis on RNA purified from GFP-p65 knock-in cells under either constant or pulsed TNF-α stimulation (*Zambrano et al., 2016*). Transcriptional outputs were subjected to unsupervised clustering of standardized profiles to group the transcription profiles by their shape, while minimizing differences in fold-changes (Materials and methods). For constantly stimulated cells, gene expression profiles reproduce well the previously observed kinetics (*Rabani et al., 2011*; *Sivriver et al., 2011*), in which the maximum expression level is achieved with different timings for different genes, and no evidence of the oscillatory dynamics of NF-κB is discerned (*Figure 6—figure supplement 1*).

We then quantified genome-wide expression profiles in the population shown in *Figure 5*, which oscillates synchronously in the forcing regime of $T_f$=180 min (Materials and methods). We found 970 genes whose expression responds to TNF-α stimulation, of which 499 increase and 471 decrease relative to unstimulated cells. Genes distributed in six highly homogeneous clusters; a higher number of clusters did not reduce significantly the inter-cluster distance (*Figure 6—figure supplement 2A*). Three of the clusters contain genes with increasing expression and three contain genes with decreasing expression (*Figure 6A*, *Figure 6—figure supplement 2*). The clusters display profiles that range from the oscillatory dynamics of IκBα (*Cluster 1*) to the slowly increasing dynamics of Ccl5 (*Cluster 3*). Notably, while genes contained in the three clusters of increasing expression are enriched with known NF-κB target genes ($10^{-14}$ < p < 0.05, Fisher's exact test, Materials and methods and *Figure 6—figure supplement 3*), clusters with decreasing expression are not significantly enriched. The same is observed in chronic stimulation (*Figure 6—figure supplement 4*).

We tested if our model would also reproduce the patterns of gene expression dynamics at a genome-wide level in these conditions, keeping the same parameters $P_S$ and $P_{NF-κB}$ as in *Figure 5* but using different gene expression parameters $P_G$ for each gene. Indeed, the model fits well the dynamical patterns observed experimentally in clusters with increasing expression (*Figure 6B*, grey lines). The median and the intervals of observed and simulated single gene expression levels match remarkably well (*Figure 6B*, red and blue lines respectively, individual fittings are shown in *Figure 6—figure supplement 6*, bottom-left panels); the average relative error per timepoint is less than 10% (*Figure 6—figure supplement 5*, left, further details on the fittings and metrics used are given in Materials and methods).

Genes in the clusters with decreasing expression cannot be fitted with the same accuracy (the average relative error per timepoint is above 10%, see *Figure 6—figure supplement 5*, right, individual gene fitting examples shown in *Figure 6—figure supplement 6*, bottom-right panels), further suggesting that these genes are not controlled directly by NF-κB. Indeed, an analysis of the parameters show that gene expression is essentially fitted for those genes as a simple RNA degradation with nearly no contribution of the gene activation/inactivation processes (low $K_{on,G}$ and $K_{off,G}$, see *Figure 5—figure supplement 1* and Materials and methods).

Interestingly, our mathematical model indicates that for a given stimulus impinging on NF-κB, the degradation rate of the mRNA is the key

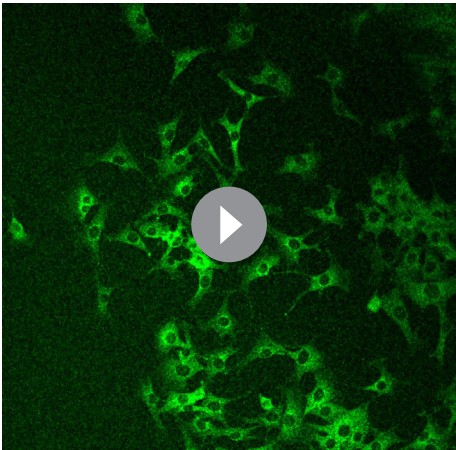

**Video 6.** Dynamics for $T_f$=60 min. GFP-p65 cells can be synchronized under periodic stimulations with $T_f$=60 min ($T_1$ =$T_2$ =30 min) when $D_1$ =10 ng/ml and $D_2$ =0 ng/ml. Twelve-hour imaging.

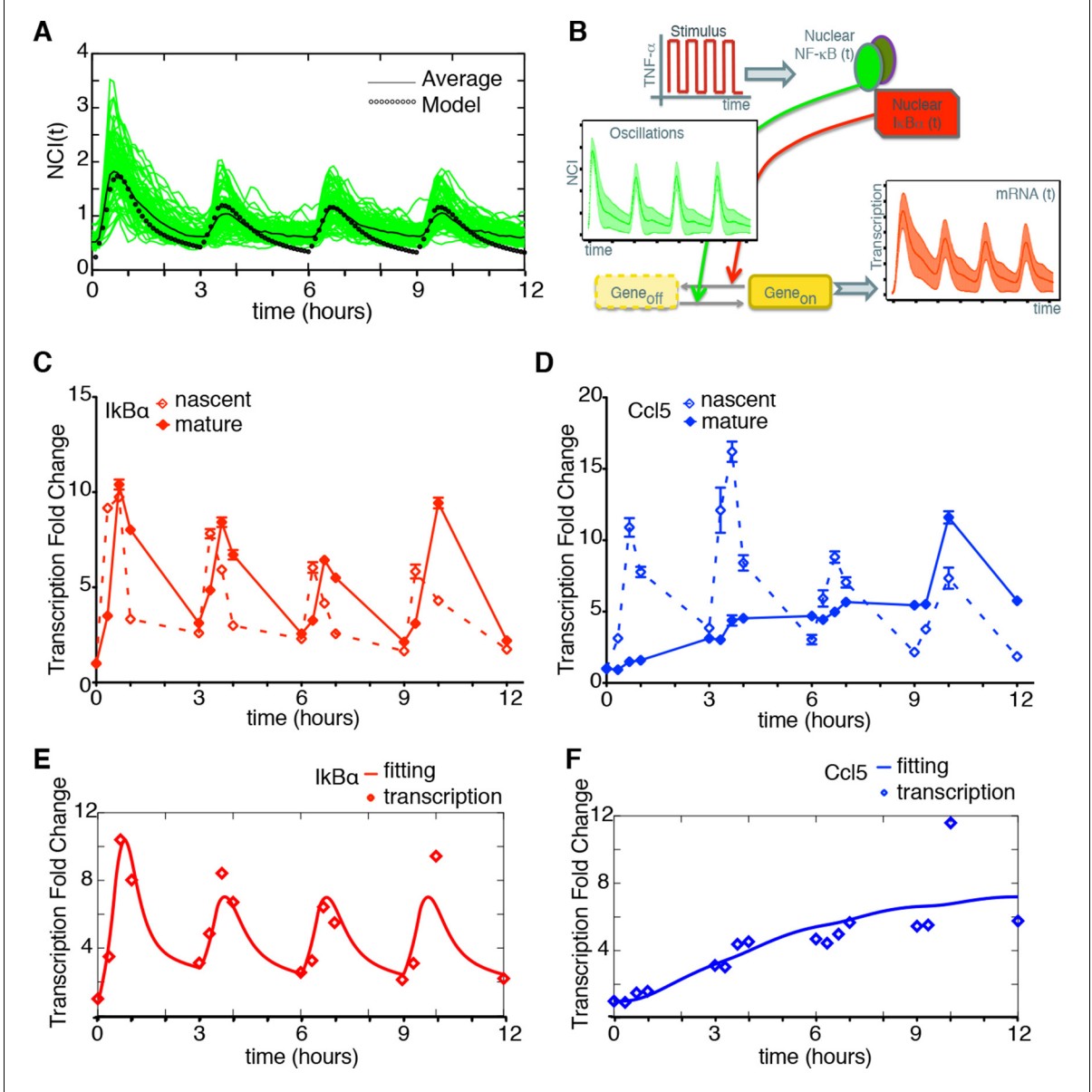

**Figure 5.** Synchronous NF-κB oscillations lead to population-level coordinated transcription. (A) NCI plot of single cell oscillations (green lines) and population average (black lines) for cells stimulated with $D_1$=10 ng/ml TNF-α, $D_2$=0 ng/ml, $T_1$=30 min and $T_2$=150 min. The open circles represent the fitting obtained using our minimal mathematical model. (B) The mathematical model predicts waves of transcription (orange plot, right) coordinated with the stimulus (red plot, top) and p65 oscillations (green plot, left). (C, D) q-PCR time course of nascent and mature mRNAs for the prototypical early and late genes IκBα and Ccl5, respectively. (E, F) Transcription profiles for mature IκBα (red) and Ccl5 (blue) RNAs (dots) can be accurately fitted (lines) by our minimal mechanistic mathematical model. The fittings were performed by keeping common the parameters regulating the external signal ($P_S$) and the dynamics ($P_{NF-κB}$) in (A), (E) and (F) but using different gene expression parameters $P_G$ for (E) and (F). Figure supplements 1 to 2 are provided.

The following figure supplements are available for figure 5:

**Figure supplement 1.** Mathematical model based on ODEs to fit single genes transcription; parameter names are reported.

**Figure supplement 2.** Hoechst staining does not affect transcription.

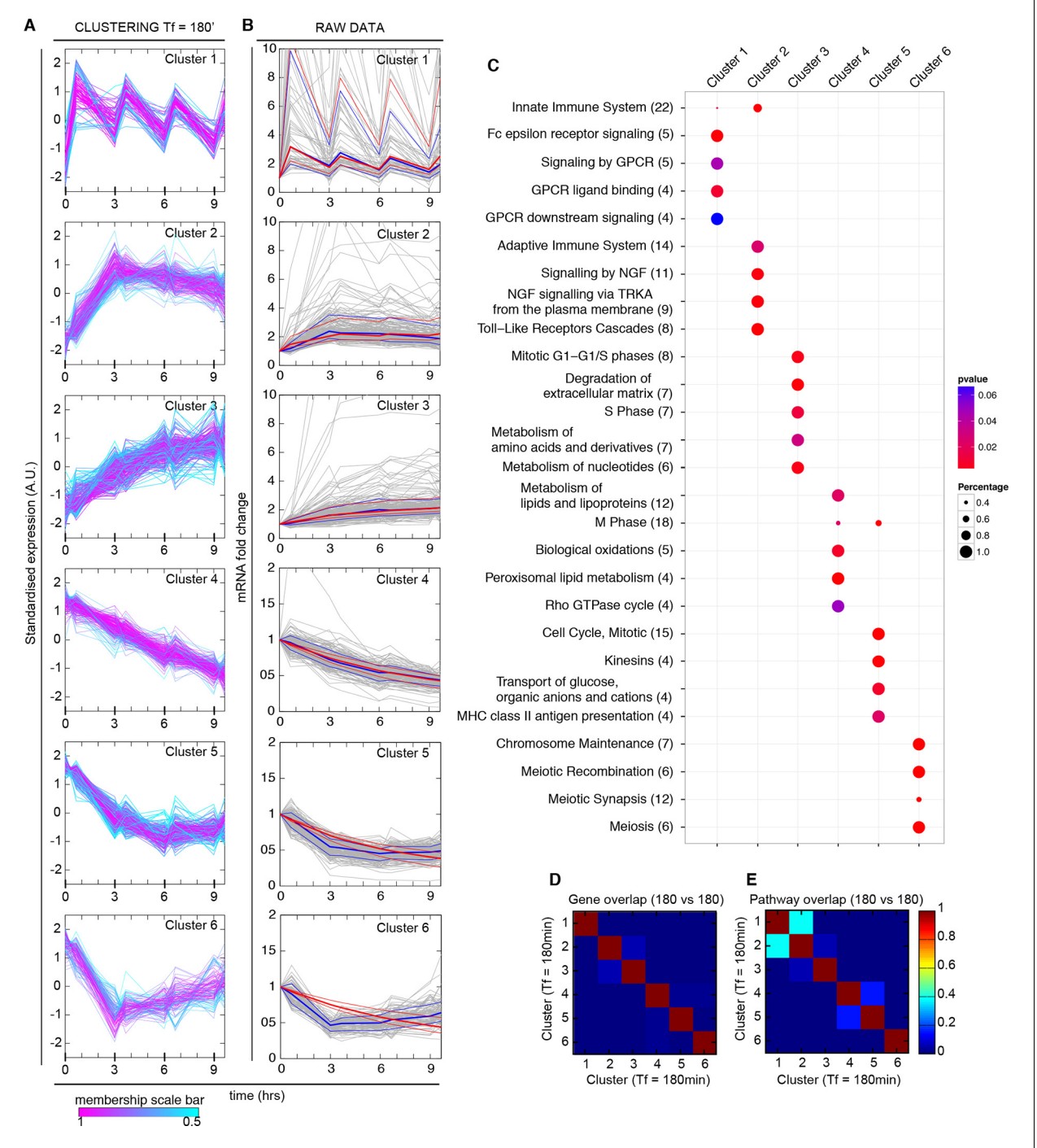

**Figure 6.** Synchronized NF-κB dynamics translates into functionally different dynamical patterns of gene expression, each corresponding to distinct pathways. (**A**) Clusters 1–6 were obtained by an unsupervised k-means-like clustering from the genome-wide transcription profiling of samples harvested in the experiment shown in *Figure 5A*. Line colours are indicative of the membership value of each gene (colour scale at the bottom). Three clusters contain genes with increasing expression (1–3) and three with decreasing expression (4–6). On the y-axis, standardized expression profile in arbitrary units (see Materials and methods). (**B**) Plots show single-gene mRNA traces (median: thick blue line; 85% and 15% intervals, thin blue lines). The time courses can also be fitted using our minimal mathematical model: shown is the median of the single-gene fits (thick red line) and the 85% and 15% intervals (thin red lines). Fittings were performed using the same parameters for the external signal ($P_S$) and the dynamics ($P_{NF-κB}$) as in *Figure 5*, but using different gene expression parameters $P_G$ for each gene. (**C**) Top five pathways of hierarchical level 2 and 3 in the Reactome database significantly enriched in each dynamical cluster. Dot sizes are proportional to the percentage of genes in the cluster belonging to that pathway. Dot

*Figure 6 continued on next page*

*Figure 6 continued*

colours identify the corresponding *p*-values (*p*-value < 0.05 is set as threshold). Scale bars on the right. (**D**, **E**) Heatmaps shows the degree of overlap at gene level (D) and pathway level (E) between each of the 6 clusters. Colour scale bar on the right. Figure supplements from 1 to 6 are provided.

The following figure supplements are available for figure 6:

**Figure supplement 1.** Transcription in cells chronically stimulated with TNF-$\alpha$.

**Figure supplement 2.** Mathematical validation of clustering.

**Figure supplement 3.** Cluster enrichment analysis for NF-$\kappa$B targets in genes clustered and displayed in *Figure 6*.

**Figure supplement 4.** Cluster enrichment analysis for NF-$\kappa$B targets in genes clustered and displayed in *Figure 7* (constant stimulation).

**Figure supplement 5.** Distribution of fitting distances.

**Figure supplement 6.** Degradation rates values are the key parameter to reproduce different gene expression patterns.

**Figure supplement 7.** Fitting of transcription data from the 180 min synchronization experiment with an alternative model of transcription.

parameter for producing the most distinct dynamical profiles. Of note the highest degradation rates are specific for oscillating genes, while those for slowly increasing genes are two orders of magnitude lower (*Figure 6—figure supplement 6*, top panels). This further underlines the importance of mature RNA processing and degradation in the definition of different patterns of gene expression.

Although the role of IκBα as a repressor has been proposed in a number of theoretical analyses (*Kellogg and Tay, 2015*; *Lipniacki et al., 2004*; *Lipniacki et al., 2006a*; *Lipniacki et al., 2006b*; *Tay et al., 2010*) only limited evidence of this role is experimentally available (*Arenzana-Seisdedos et al., 1995*). Thus, to investigate the role of IκBα in patterning gene expression, we built a model where IκBα does not act as a direct repressor (*Figure 6—figure supplement 7A,B*). This alternative model fits the gene expression dynamics of each cluster well (*Figure 6—figure supplement 7E*), and again better for genes with increasing transcription than for those decreasing (*Figure 6—figure supplement 7C*). The mRNA degradation rate is again the key parameter to discriminate between the different patterns of gene expression (*Figure 6—figure supplement 7D*). While our fittings do not provide conclusive evidence on the role of IκBα as a transcriptional repressor, they suggest that this is not needed.

Finally, to understand if the different dynamics observed with $T_f$=90 min were also present in other conditions, we used the same clustering approach for gene expression profiles of cells stimulated with $T_f$=90 min, $D_1$ =10 ng/ml and $D_2$ = 0 ng/ml (as in *Figure 1C*). Oscillations in transcript levels were observed, although they were not as conspicuous as for $T_f$=180 min (*Figure 7A*), probably due to the fact that 90 min is not long enough for most transcripts to degrade. Enrichment of target genes is similar to the two previous cases (*Figure 7C,D*).

We then performed a pathway analysis to determine if each of our dynamical clusters were enriched in functionally related pathways (see Materials and methods). Indeed, this is the case (*Figure 6C*): *Cluster 1* is broadly related to chemokines and chemokine receptors and contains so-called early genes, *Cluster 2* to the immune system and so-called intermediate genes, and *Cluster 3* to extracellular matrix rearrangements and so-called late genes (*Rabani et al., 2011*; *Tian et al., 2005a*; *2005b*). Thus, the clustering also indicates a precedence order in the articulation of the cell's response. Genes with decreasing expression comprise pathways related to metabolism (*Cluster 4*), cell cycle (*Cluster 5*) and chromosome maintenance (*Cluster 6*).

The overlap between clusters is minimal both at gene and pathway level (*Figure 6D* and *Figure 6E*, respectively, and Materials and methods). The correspondence between dynamics and gene function is also fulfilled in cells forced with $T_f$=90 min (*Figure 7B*), with a low intercluster overlap at gene level (*Figure 7E*), and slightly higher overlap at pathway level (*Figure 7F*). Interestingly, there is a good correspondence between clusters identified upon both 180 and 90 min forcing. In

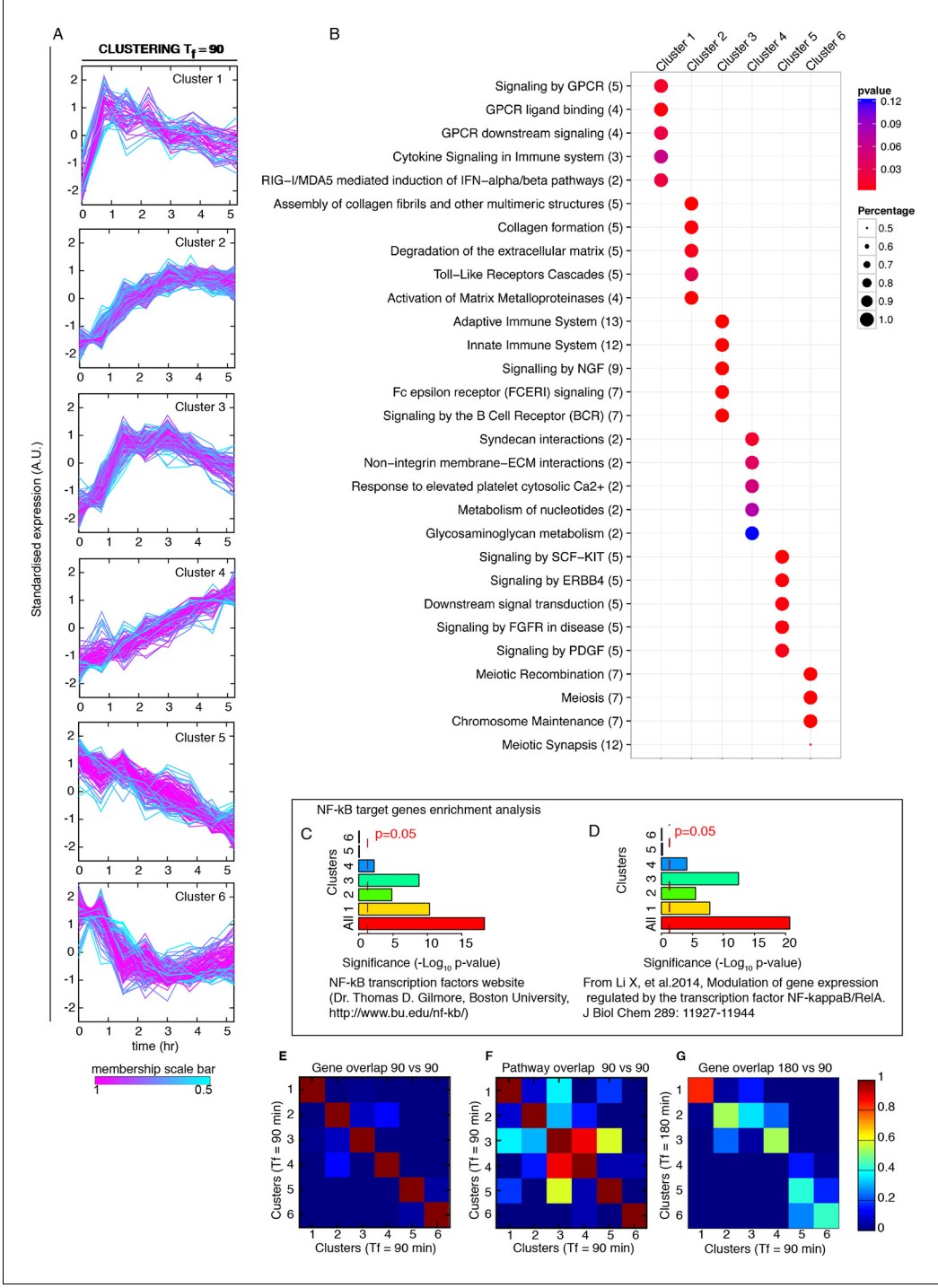

**Figure 7.** Genome-wide clustering for $T_f$=90 min. (**A**) Clusters obtained by unsupervised k-means clustering from the genome-wide transcription profiling of cells perturbed with a forcing of 90 min, $D_1$=10 and $D_2$=0 ng/ml of TNF-α. Lines colours: blue and red indicate low and high membership values, respectively. (**B**) Top five pathways of hierarchical level 2 and 3 in the Reactome database significantly enriched in each dynamical cluster. Dot sizes are proportional to the percentage of genes in the cluster belonging to that pathway. Dot colours identify the corresponding *p*-values (*p*-value<0.05 is set as threshold). Scale bars on the right. (**C, D**) Enrichment analysis for NF-κB targets from the clusters displayed in panel A. Two lists of NF-κB targets were considered: left: Gilmore's web-site (www.bu.edu/nf-kb/); right: data from Brasier and Kudlicki groups (*Li et al., 2014*). Significance is shown as -Log(p-value), a dashed line marks the threshold of significance at p=0.05. (**E, F**) Heatmaps show the degree of overlap at gene level (**E**) and pathway level (**F**) between each of the 6 clusters. (**G**) Overlap at a gene level between

*Figure 7 continued on next page*

*Figure 7 continued*

clusters obtained for 180 min and for 90 min. It is particularly high for Clusters 1, those of oscillating genes. Colour scale bar on the right. Figure supplement 1 is provided.

The following figure supplement is available for figure 7:

**Figure supplement 1.** Synchronous NF-κB oscillations arising with 90 min forcing lead to population-level coordinated transcription.

particular *Cluster 1* (the oscillating cluster) contains almost exactly the same genes in both conditions (*Figure 7G*).

Taken together, the above results indicate that different gene expression dynamics identify functionally related categories of genes, suggesting a strict relationship between dynamics and function in the cell's response to external stimuli.

## Discussion

### The NF-κB system synchronizes to external periodic forcing as a damped oscillator

Our results show that single cells activate NF-κB signalling synchronously to external TNF-α stimulation, and that repeated forcing elicits synchronous NF-κB oscillations at population level. However, NF-κB does not behave as a free oscillator: no cell oscillates constantly in the presence of continuous stimulation (*Figure 8A and B* left, green lines). Furthermore, synchronization among cells does not improve upon repeated forcing ("training"), and single cells trained for a dozen forcing cycles stop oscillating and dephase ("forget") as fast as cells stimulated only once. Finally, cells can be synchronized with a wide variety of forcing periods and forcing amplitudes, and synchronize to the external forcing (*Figure 8A and B* right, green lines) without showing any preference for periods resonating with the NF-κB intrinsic period of 90 min; synchronization is more pronounced with high than with low stimulation amplitude.

The ability of NF-κB to oscillate in tune with a forcing of 45 min illustrates well the plasticity of synchronization: 45 min is equivalent to one half of the intrinsic period, and entrained cells would skip one forcing period out of 2 (following the 1:2 resonance), but our data do not support this interpretation, since cells only infrequently skip a forcing period. Simulations with our mathematical model can actually reproduce the ability of the system to synchronize to different forcing periods. The ability of our cells to synchronize to a 60 min periodic forcing points in the same direction.

Thus, we find that our GFP-p65 knock-in cells are not entrained, as do not satisfy the essential precondition – sustained oscillations – and two critical tests for entrainment – increasing synchronization upon repeated forcing, and synchronization to the natural frequency of the internal oscillator (*Pikovsky et al., 2003*). Our cells appear closer to the classical textbook example of a damped harmonic oscillator (*Goldstein et al., 2001*), whose frequency corresponds to the frequency of the external stimulation, and whose synchronization to the external stimulus increase monotonically with stimulus amplitude.

We interpret the damped oscillator behaviour of the NF-κB system as an intrinsic characteristic of the NF-κB system which allows it to adapt to a wide variety of inputs and to quickly reset. In fact, pathogens and inflammatory signals do not come in regular periodic patterns, and a carefully calibrated inflammatory response that would adapt itself to any irregular pattern of stimulation would be positively selected for.

We explored our minimal mathematical model to understand why we observed damped oscillations whereas Kellogg and Tay observed sustained oscillations under constant stimulation, a behaviour that presumably is the root of the different synchronization mechanisms observed. Our simulations suggest that sustained oscillations would not be the norm, but rather the result of a specific set of parameters (see *Figure 1—figure supplement 5A,B*). Indeed, the NF-κB network has an equilibrium that depending on the combinations of parameters can be stable, giving rise to damped oscillations that converge to it, or unstable, giving rise to a stable cycle around it (see e.g.

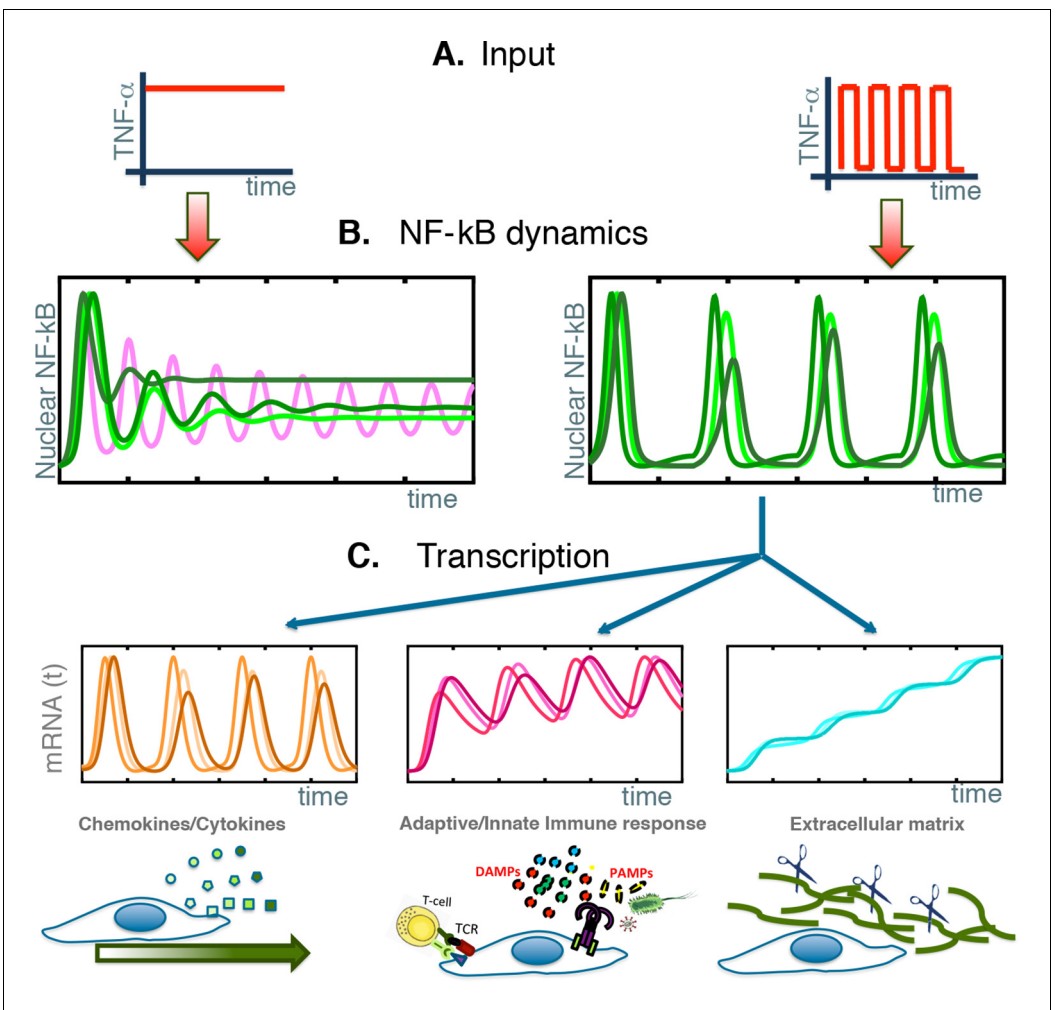

**Figure 8.** NF-κB behaves as a damped oscillator that can synchronize to time-varying external stimuli to produce functionally related transcriptional outputs. (**A**) The NF-κB system is able to provide different responses to different inputs, from constant (left) to time-varying ones (right). (**B**) Our cells show damped oscillations to a constant stimulus (left, green lines), although for other cell types sustained oscillations might be possible (magenta line). Damped oscillations can adapt to timevarying inputs (right) and give rise to synchronous oscillations. (**C**) These synchronous oscillations produce different patterns of gene expression, from oscillating (left, orange lines) to slowly increasing (right, blue lines) and intermediate dynamics (pink lines, centre). We find that each kind of dynamics is typical for genes involved in different cellular functions.

*Figure 8B*, left, magenta line); transitions between these two states can be mediated by a Hopf bifurcation.

The simulations performed with our simple model can thus reconcile our experimental observations with the detection of sustained oscillations for different cells (*Kellogg and Tay, 2015*), but also with the rich variety of damped oscillatory dynamics reported for other cell types in recent works (*Cheng et al., 2015*; *Sung et al., 2014*). Other theoretical analyses have identified the amount of NF-κB expressed by a specific cell as the determinant of the Hopf bifurcation and the stability of the oscillator (*Mothes et al., 2015*). Notably, we used cells with physiological and uniform p65 levels. We also note that the interplay between different feedback loops, including the A20 feedback (*Werner et al., 2008*) can vary between cell types, leading to different degrees of dampening, as observed experimentally by selective knock-outs (*Hoffmann et al., 2002*); dissecting the contribution of each of these feedbacks will be the subject of future experimental and theoretical work.

## Oscillations in the nuclear concentration of NF-κB lead to different dynamical patterns of gene expression

The oscillatory behaviour of NF-κB consists in periodic relocalization between the nucleus and the cytoplasm, and entails corresponding cycles of NF-κB binding to cognate binding sites in the genome. Thus, NF-κB driven transcription should be pulsatile as well, and in fact we detect cycles in the levels of elongating transcripts in the prototypical early and late NF-κB-controlled genes IkBα and Ccl5. However, whereas mature mRNA oscillates for IκBα, it slowly accumulates for the late gene Ccl5, and this held true for different frequencies of the periodic stimulation. Genome-wide analysis of dynamics, using unsupervised clustering of whole transcriptome profiles, identified six highly homogeneous sets ("clusters"), three containing genes with increasing transcription and three containing genes with decreasing transcription. While the genes with increasing transcription are predicted NF-κB targets, those in the decreasing sets are not.

The train of p65 nuclear translocations is thus decoded by the cell into different transcriptional dynamics: genes in Cluster 1 display a clear and sustained oscillation in transcript level; instead, transcripts for genes in Clusters 2 and 3 accumulate fast or increase slowly and steadily over many cycles of NF-κB oscillations, respectively (see *Figure 8C*). The result is that each new wave of TNF-α elicits on Cluster 1 genes almost the same responses as the previous waves, and no memory is kept of the past; the expression of genes of Clusters 2 and 3 allows a long-lasting response that integrates over time the responses to previous waves.

All three dynamics of genes with increased expression could be reproduced by our minimal mathematical model of transcription, which incorporates the features of the paradigmatic telegraph model (*Suter et al., 2011*). Our model also suggests that mRNA degradation is the key parameter that allows a functional and temporal shaping of the NF-κB response, in particular for transcripts in Cluster 1 (*Figure 6—figure supplement 6*, top-left panel). The computational results fit well with the fast turnover of early transcripts, whose degradation depends on positive regulators like Zfp36 (*Rabani et al., 2011*). Of note, Zfp36 is in Cluster 1, and oscillates sharply and synchronously with the forcing; this might contribute further to the shaping of peaked responses.

## Oscillatory dynamics and functionally related patterns of gene expression: an adaptive viewpoint

Overall, our most remarkable finding is that NF-κB oscillations drive the expression of distinct sets of genes whose expression dynamics and functions correlate: the transcripts that oscillate encode mostly cytokines and cytokine receptors, the ones that rise fast and decrease slowly encode proteins involved in immunity, and the ones that rise steadily encode proteins involved in the rearrangement of the extracellular matrix (*Figure 8C*).

While steady-rise and rise/decline programs of gene expression can be operated by non-oscillatory transcription factors, oscillating transcription programs can be best achieved with oscillating transcription factors. We note that the transcripts that oscillate are involved in the decision of cells of whether to move or not, which has to be refreshed repeatedly over time. The oscillatory dynamics of NF-κB allows time to be segmented in units of 90 min cycles, with the option to resume chemokine-directed migration after each single cycle. As seen from this perspective, the operation of the NF-κB system as a damped, fast-detuning oscillator implies that in the presence of constant stimulation each motile cell will decide for itself, without synchrony to nearby cells. In contrast, in a situation where stimuli change over time beyond a certain threshold – at least three-fold variation in amplitude – cells would be broadly coordinated. Uncoordinated movement may maximize the volume randomly patrolled by cells, while coordinated movement may be optimal to reach a specific target location.

We thus speculate that NF-κB oscillations may have adaptive value in achieving oscillatory expression of genes involved in movement and direction, and the cells' decision between uncoordinated or collective action. In conclusion, we suggest that the oscillatory dynamics of NF-κB (and other transcription factors) is a means of segmenting time to provide opportunity windows for decision.

## Materials and methods

Please note that this section contains both technical details and supplemental information explaining the rationale behind our experimental approaches. We also refer here to Figure supplements that are also discussed in the main text.

### Cell lines and cell culture

GFP-p65 knock-in fibroblasts were provided by M. Pasparakis (details in *De Lorenzi et al (2009)*; *Sung et al (2009)*) and cultured in phenol-red free DMEM, with 10% FCS, 50 μM β-mercaptoethanol, 1x L-glutamine, 1x sodium pyruvate, 1x non-essential amino acids, and 1x pen/strep in standard tissue culture plastic.

GFP-p65 fibroblast cultures were started from original aliquots frozen upon arrival from Pasparakis' lab and tested every week to exclude mycoplasma contamination using approved kits. Since there are no approved STR profiling protocols for mouse cells ((*Yu et al., 2015*) and http://www.atcc.org/Global/FAQs/0/A/STR%20Testing%20Service%20-%20STR%20on%20mouse%20cell%20lines.aspx), we carefully tested our cells for the presence of non-green cells (not expressing GFP-p65) that could represent a cross-contamination of the original culture.

For imaging experiments, cells were plated one day before the experiment in CellASIC™ ONIX M04S-03 Microfluidic Plates at low density to avoid confluence on the day of the experiment (e.g. *Figure 1—figure supplement 1A*). These plates consist of chambers for cell culture connected through microfluidic channels to a series of reservoirs containing media with selected concentrations of stimuli that can be flown through the chambers. Of note, to avoid cell stress or toxicity, the microfluidic plates are primed with 10%FCS in DMEM for 2–4 hr before cell plating.

Before imaging, DMEM-0.1% FCS medium containing 50 ng/ml of the nuclear dye Hoechst33342 was replaced in the microfluidic chambers 3 hr prior to the experiment using the microfluidic platform, see details on the use of the CellASIC™ ONIX below. Mouse recombinant TNF-α (R&D Systems) was diluted in DMEM-0.1%FCS as specified in the text and added in the plate reservoirs. It is relevant to note that TNF-α activity is maintained after 12 hr incubation in the plate reservoirs at 37°C, indicating that TNF-α degradation is negligible when it is not in contact with the cells (*Figure 1—figure supplement 1D*).

### Microfluidics

The CellASIC® ONIX Microfluidic Platform allows to apply sharp stimuli by quickly replacing the medium in the cell chambers. The flexible proprietary software allows the delivery of medium containing different concentrations of TNF-α for specific time sequences for more than 10 hr while cell are imaged. The protocols are available upon request. The medium flows in the channels around the microfluidic chambers and diffuses through the perfusion barrier, minimizing the undesirable effect of shear stress (*Babini et al., 2015*) (see *Figure 1—figure supplement 1A,B*). The small volume of medium in the chambers (less than 1 μl) is replaced fast even at low pressures, as confirmed by the sharp oscillatory profiles obtained even for high frequency stimuli. In spite of the low pressure applied (1 psi), the flow rates of 10 μl/hr (manufacturer's website) are in the same order of magnitude as the 200 nl/min used in *Kellogg and Tay (2015)*. We also generated "sawtooth-like" profiles as in *Kellogg and Tay (2015)* by periodically replacing the medium with a 15 min pulse and stopping the flow. However, the cell-to-medium ratio in our culture chambers might be different and relevant for TNF clearance due to cell-linked decay needed for the sawtooth profile.

### Live imaging and detection of NF-κB dynamics

Live cell imaging of GFP-p65 knock-in MEFs was performed using a Leica TCS SP5 confocal microscope with an incubation system where cells were stably maintained at 37°C in 5% CO2. Time-lapse images were acquired at 6 min intervals for more than 15 hr. We used a low magnification objective (20x, 0.5 NA) and an open pinhole (Airy 3), ensuring that the image depth (10.7 μm) contains the thickness of the whole cell so that images are a record of the total cell fluorescence. GFP-p65 is imaged with the 488 nm Argon laser (GFP channel) while Hoechst 33342 stained nuclei are imaged with the low energy 405 nm UV diode laser at 5% of its maximum intensity (HOE channel). Images were acquired as 16 bit, 1024×1024, TIFF files. Experiment replicates were acquired on different days starting from different batches of frozen cells (samples provided as *Videos 1–5*).

Nuclei were stained with Hoechst 33342 for imaging, segmentation and tracking (*Zambrano et al., 2014a*) without any interference with the natural signalling dynamics (see **below** for a description of the controls performed).

## Safety assessment for Hoechst staining

Stress from photo-damage during live-cell imaging can hamper cell biology studies (*Cole, 2014*), while nuclear dyes can interfere with the natural signaling dynamics (*Ge et al., 2013*; *Martin et al., 2005*). Hence different controls were performed to exclude distortions of the natural signalling in our imaging conditions.

### Apoptosis and cell death

We noticed that cells under a constant flow of fresh medium do not display apoptotic events until very late in the experiments (10–12 hr; *Video 2*). Importantly, unstimulated cells are viable and undergo mitosis despite the presence of low serum concentration (0.1% FCS). In contrast, cells receiving a constant flow of TNF in the absence of Hoechst and UV imaging undergo apoptosis (*Video 1*), with a frequency that is proportional to the TNF concentration (*Figure 1—figure supplement 7*). The imaging fields contain detectable apoptotic cells after 3, between 6 and 9 or after 12 hr of stimulation with 10, 1 or 0.1 ng/ml of TNF, respectively. This suggests that cell death observed in *Video 1* was due to the continuous flow of 10 ng/ml TNF-α and not due to the imaging conditions.

### Photo-damage

The cells were either imaged for 15 hr in the presence of Hoechst or left without Hoechst and not imaged; we then immunostained cells to detect thymine dimers formation. Immunostaining intensities (*Figure 1—figure supplement 6*) are extremely low in cells exposed to Hoechst and imaging, especially when compared to cells exposed to UVC radiation. This suggests that imaging does not produce thymine dimers or that thymine dimers are repaired fast and do not accumulate in the cells.

To further exclude the possibility of DNA damage, we checked for the presence of gammaH2AX, which is a marker for early DNA repair after formation of DNA double strand breaks (DSB) (*Turinetto and Giachino, 2015*; *Yang et al., 2015*) and late DNA repair after the formation of UV-induced thymine dimers (*Oh et al., 2011*; *Staszewski et al., 2008*).

Again, the signal for gammaH2AX (*Figure 1—figure supplement 7*) is extremely low in imaged cells, when compared to the signal from cells treated with doxorubicin, which induces DSBs. Of note, the phosphorylated form of H2AX is found at DNA polymerase stalled forks (*Turinetto and Giachino, 2015*) as marker of replicative stress. This observation may explain the minimal signal present in some of the imaged cells. All together these results indicate negligible DNA damage in the conditions considered.

### Interference with NF-κB dynamics

We performed a manual tracking of cells that received a constant flow of TNF-α but were not exposed to Hoechst nor to UV light ("unstained/non-imaged cells", *Figure 1—figure supplement 9*). Manual segmentation of nuclei is simple when nuclei are either empty or full of p65, but becomes unreliable when concentrations in nucleus and cytoplasm are similar. Despite this caveat, we found a good qualitative agreement of dynamics in unstained/non-imaged cells with the heterogeneous but damped oscillatory dynamics observed under Hoechst/UV imaging conditions, as shown in *Figure 1—figure supplement 9*. Therefore, the p65 dynamics described in our manuscript represents the biological effect of stimulation with TNF-α, and is essentially undistorted by imaging.

## Immunobloting and immunostaining

Staining was performed according to manufacturers' instructions. For WB, we used an anti p65 Rab (dil 1:1000, #sc-372 C20, Santa Cruz; *Figure 1—figure supplement 10*). For immunostaining we used anti gammaH2AX Mab (dil. 1:500; #05-636, Millipore; *Figure 1—figure supplement 7*) and anti thymine dimers TDM-2 (dil. 1:2000, Cosmo Bio, Japan (*Komatsu et al., 1997*); *Figure 1—figure supplement 6*). UV-photodamage was induced with 254 nm UV irradiation using an UVC 500

Crosslinker, Amersham. Panels in *Figure 1—figure supplements 6 and 7* are representative of 3 independent experiments.

## Quantification of NF-κB dynamics

The following short description summarises the whole process in few sentences. Each passage will be then thoroughly described in the next paragraphs.

The dynamics were quantified by computing the nuclear to cytoplasmic ratio of the intensities (NCI) of single cells, which is a measure robust against distortions and reflects faithfully the oscillations in the nuclear concentration of NF-κB oscillatory dynamics. This holds true provided that the total amount of p65 is constant, as we show for our cells under stimulation (see *Figure 1—figure supplement 9*), and is also assumed by mathematical models (see *Hoffmann et al (2002)* and the more recent in *Kellogg and Tay (2015)*). Interestingly, p65 is not constant in macrophages stimulated with LPS (*Sung et al., 2014*). The analysis of time series relies on the detection of significant peaks, performed following our previously discussed procedure (*Zambrano et al., 2014a*) that allows to distinguish significant peaks from noisy peaks. Once peaks are detected we assign them a phase: $2\pi$ for the "maxima" and $\pi$ for the "minima" between peaks.

## Selection of the quantifier of NF-κB dynamics

In order to assess the oscillations for a larger number of cells, we optimized our software (*Zambrano et al., 2014a*) to calculate the nuclear to cytoplasmic ratio of the intensity (NCI) of NF-κB signal for hundreds of cells; NCI is a quantifier that has been used by other groups (*Ashall et al., 2009*; *Nelson et al., 2004*). As we argue below, thanks to the fact that the total amount of NF-κB is constant (*Figure 1—figure supplement 10*), NCI depends univocally and monotonically on the nuclear amount of NF-κB. More importantly, due to the fact that it is a ratio of intensities, it is robust and independent of slight changes in the focus and in the laser intensity, among other possible experimental distortions, which are observed in our setup. The same rationale brought us to use ratios of intensities in our previous works (*Sung et al., 2009*; *Zambrano et al., 2014b*). Importantly, these distortions imply that the background-adjusted mean nuclear intensity used in other studies (*Kellogg and Tay, 2015*; *Lee et al., 2014*; *Sung et al., 2014*), although advantageous for other reasons (it only requires the segmentation of the nuclei and estimation of the background) would not be appropriate in our imaging experimental setup.

We provide below a more detailed argumentation of these ideas.

Following the notation of *Zambrano et al (2014b)* we have that the intensity measured in pixel *p* of the image at time *t* in a time lapse experiment can be described as:

$$I(p,t) = A(p,t)P(p,t) + B(p,t) \tag{Q1}$$

where *P(p,t)* is the amount of NF-κB in pixel *p*, *A(p,t)* is the amplification coefficient between the protein brightness and the amount of protein and *B(p,t)* corresponds to the background. In our experiments with time-varying intensities, it is clear that both *A(p,t)* and *B(p,t)* vary in time, and presumably also in space, due to laser variations and/or slight variations in the illumination uniformity.

Our quantifier NCI, that we estimate using our software, is the ratio of the background corrected average intensities in the nucleus and the cytoplasm $\left(\langle I(t)\rangle_{nuc} \text{ and } \langle I(t)\rangle_{cyto}\right)$ respectively, and can be defined using this notation as:

$$NCI(t) \equiv \frac{\langle I(t)\rangle_{nuc}}{\langle I(t)\rangle_{cyto}} = \frac{S_{cyto}}{S_{nucleus}} \frac{\sum_{p\in nucleus} I(p,t) - B(p,t)}{\sum_{q\in cytoplasm} I(q,t) - B(q,t)} = \frac{S_{cyto}}{S_{nucleus}} \frac{\sum_{p\in nucleus} A(p,t)P(p,t)}{\sum_{q\in cytoplasm} A(q,t)P(q,t)} \tag{Q2}$$

so we have that, if we consider that *A(p,t)* is approximately constant and equal to certain *A(t)* in the area occupied by the cell (as it is in our images):

$$NCI(t) \cong \frac{S_{cyto}}{S_{nucleus}} \frac{A(t)\sum_{p\in nucleus} P(p,t)}{A(t)\sum_{p\in cyto} P(q,t)} = \frac{S_{cyto}}{S_{nucleus}} \frac{(NF - \kappa B_{nuc}(t))}{(NF - \kappa B_{cyto}(t))} \tag{Q3}$$

where $S_{cyto}$ and $S_{nuc}$ are the areas occupied by cytoplasm and the nucleus, respectively. Our quantifier is hence a good indicator of the ratio of the amount of NF-κB in the nucleus and in the

cytoplasm. However the numerator and denominator of this ratio can fluctuate due to biological reasons and this might blur the existence of oscillations in the nuclear amount of NF-κB. But this is not the case, due to the fact that the total amount of NF-κB is constant for our stimulated fibroblasts. This has been assumed in different mathematical models present in the literature, from the seminal paper (*Hoffmann et al., 2002*) to more recent papers (*Kellogg and Tay, 2015*). Interestingly, though, this view has been challenged by the recent work of *Sung et al. (2014)*, which shows that in macrophages under LPS stimulation p65 is regulated by a positive feedback. Thus, to confirm that our assumption is reasonable, we quantified p65 by western blotting for our cells under 10 ng/ml TNF-α at several timepoints from 1 to 8 hr. The results are shown in (*Figure 1—figure supplement 10*), showing no significant change in the p65 levels upon stimulation. Hence, we consider our assumption valid. This implies that:

$$NCI(t) = \frac{S_{cyto}}{S_{nucleus}} \frac{(NF-\kappa B_{nuc}(t))}{(NF-\kappa B_{TOT})-(NF-\kappa B_{nuc}(t))} \tag{Q4}$$

hence we can see that *NCI(t)* depends monotonously on $(NF-\kappa B_{nuc}(t))$, see *Figure 1—figure supplement 9B*. For this reason, it is intuitively clear that each local maximum or minimum of $(NF-\kappa B_{nuc}(t))$ leads to a local maximum or minimum of $NCI(t)$. Hence, oscillations in the nuclear amount of NF-κB will be observed also using *NCI(t)*. We can put this mathematically by saying that oscillations in the nuclear concentration of NF-κB occur at times $t$ for which the condition $(NF-\kappa B_{nuc}(t))' = 0$. Similarly, oscillations in NCI will depend on the value of the derivative of NCI, that is:

$$NCI'(t) \propto \frac{((NF-\kappa B_{TOT})-(NF-\kappa B_{nuc}(t))+1)}{((NF-\kappa B_{TOT})-(NF-\kappa B_{nuc}(t)))^2}(NF-\kappa B_{nuc}(t))' \tag{Q5}$$

so from the above formula it is easy to see that:

$$NCI'(t) = 0 \longleftrightarrow (NF-\kappa B_{nuc}(t))' = 0 \tag{Q6}$$

Overall, then, imaging, mathematical and biochemical arguments confirm that computing $NCI(t)$ is an adequate way to quantify oscillatory dynamics.

Concerning the background-corrected average intensity, using the above notation it would be calculated as:

$$\langle I(t) \rangle_{nucleus} = \frac{1}{S_{nucleus}} \sum_{p \in nucleus} I(p,t) - B(p,t) = \frac{1}{S_{nucleus}} \sum_{p \in nucleus} A(p,t)P(p,t) \tag{Q7}$$

thus $\langle I(t) \rangle_{nucleus} \propto \sum_{p \in nucleus} P(p,t) = (NF-\kappa B_{nuc}(t))$ for all $t$ only if $A(p,t)$ were constant in all time frames. As argued previously, this is not the case in our setting and that's the reason why we preferred to use NCI.

Finally, variations in the areas of the nucleus and the cytoplasm might introduce small distortions, although they typically vary seldom and slowly. Notice though that our software discards cells for which the areas change abruptly, which typically is an indicator of imminent mitosis or cell death. We cannot totally exclude variations of p65 at single cell level, provided that we measured it using a population assay; however, p65 turnover is known to be long, so its slow variation would contribute with a small term in the derivative and thus would just slightly distort the times for which peaks in NCI appear compared to those of the nuclear concentration. To conclude, we think that NCI would still capture the peaks and hence would be able to assess the cell's oscillatory state, which is our aim here.

The data provided confirm the validity of NCI as a quantifier. First, we have that NCI(t) remains reasonably constant – except for some spontaneous oscillations, as previously reported (*Zambrano et al., 2014b*) for cells that are not stimulated, as shown in *Figure 1—figure supplement 9A*. This is remarkable because the movie from which these series were obtained present considerable variation in the image intensity (see the *Video 2*). Along the same lines we do not find upwards or downwards average trends in our data of NCI time series for different conditions discussed in the main figures and in the figure supplements, which indicates further that the time series are properly normalized and reflect well the dynamics. As an additional confirmation, we have plotted together the average nuclear intensity and the NCI values obtained from manual segmentation

in *Figure 1—figure supplement 9C*, that show the same kind of qualitative behavior as NCI but with different trends, as expected. A last indicator of the sensitivity of the quantifier comes from its ability to detect even small oscillatory peaks, see *Figure 3* and *Figure 3—figure supplement 1* as an example. Overall, then, NCI is a faithful measure of the oscillatory state of our cells.

## Automated analysis of time-lapse experiments data

The major advantage of considering NCI with respect to the nuclear to total ratio used in *Zambrano et al (2014a)* is that it does not require a perfect segmentation of the cytoplasm, which might be complicated when cells touch each other. We developed a software that uses this quantifier and is thus able to multiply by a factor of 2 the number of cells tracked, and by a factor of 1.5 the average tracking time with respect to the one described in *Zambrano et al (2014a)*.

The software used to calculate NCI is provided as source code and works as follows: for each time-lapse experiment, we have N frames images in the HOE channel and in the GFP channel, respectively. Nuclei were segmented and used for cell tracking following the procedure described in *Zambrano et al (2014a)*. In order to estimate the average cytoplasmic intensity, the background was computed by taking a square area centred on the cell nucleus, dividing it in tiles and using the one with the smallest average intensity in the GFP channel to estimate the background intensity (this procedure gives values compatible with the values obtained using a clustering-based algorithm). Points belonging to the cytoplasm are those around the nucleus in a size window equal to 1.5 times the size of the nucleus. The average cytoplasmic intensity is computed from the intensity in the GFP channel of the pixels from this "ring". An example is shown in *Figure 1—figure supplement 1D*.

Dividing or apoptotic cells were identified assessing their geometrical features (abrupt changes in size of the nucleus and of the "cytoplasmic ring") and discarded automatically.

## Time series analysis

To analyse the resulting NCI time series, we adapted our peak detection algorithm (*Zambrano et al., 2014a*). We consider peaks as a sequence of a local minimum, local maximum and local minimum. The value of the peak θ is defined by the height of the peak (from the highest minimum to the maximum). A peak defined by only three consecutive time points is considered a noise peak. We plot the distribution of the noise peak values obtained from our unperturbed and chronic stimulation time series (*Figure 1—figure supplement 2A*), for which we observe our previously reported spontaneous activations (*Zambrano et al., 2014a*). We find that these noisy peaks have a low value, and hence by considering as significant those with a value over θ>0.15, we find a reasonably good compromise between the need to ignore noise peaks and the need to detect small peaks of valuable dynamical information. This was tested in a number of time series, and an example of time series with the significant peaks detected using this threshold is shown in *Figure 1—figure supplement 2B*. Calculations of magnitudes inferred from peaks take advantage of time series from cells that were tracked for at least 7 hr.

Following (*Mondragon-Palomino et al., 2011*) we used the peaks to assign a phase value: peak maxima are assigned a phase 0 (2π) and a phase π is assigned to peak minima. This was of particular interest to obtain an assessment of the oscillatory modes present in our system, provided that peaks can be very heterogeneous and plotting the peak height in colour-plots might make it difficult to appreciate the smaller ones and hence the possible resonant oscillatory patterns. An example of this transformation is shown in *Figure 1—figure supplement 2C*, which is the phase derived from the peaks obtained of the time series displayed in *Figure 1—figure supplement 2B*. As in *Mondragon-Palomino et al (2011)* we use the phase difference between the time series and the forcing to quantify the degree of synchrony. The phase difference Δφ was calculated from the timing ΔT between the beginning of each forcing cycle and the closest significant peak, as $\Delta\varphi = 2\pi\Delta T/T_f$, where $T_f$ is the forcing period. The entropy of the distribution of the phase is calculated as:

$$S = -\sum_k p_k \log p_k$$

where $p_k$ is the probability of the kth bin (we use eight bins for all the conditions considered). The synchrony intensity is then computed as:

$$\eta = 1 - S/S_{max}$$

where $S_{max}$ is the value obtained for equal values of $p_k$.

## Quantitative real-time PCR

Total RNA was isolated using the IllustraRNAspin Mini kit (GE Healthcare), and complementary DNA (cDNA) was obtained by retro-transcription with Random Hexamers and SuperScript II Reverse transcriptase (Invitrogen) following the manufacturers' instructions. Primers for the detection of mature transcripts were designed in adjacent exons and spanned the intervening intron; primers for nascent transcripts were located across an exon-intron boundary. Primer sequences are listed in *Supplementary file 1*.

Quantitative real-time PCR was performed with SYBR Green I protocol using the LightCycler480 (Roche). The results are shown as averages of three technical replicates. Analysis was performed with the $-\Delta$Ct method corrected for primer efficiencies (*Vandesompele et al., 2002*) and normalised with two reference genes (Actb and Rplp1) (*Nordgard et al., 2006*). Experiments were repeated twice.

It is important to emphasize that we did not stain our cells nor did we illuminate them with our UV laser in our transcription experiments, neither in the RT-PCR assays nor in the Microarray experiments described below. The controls described previously show that the nuclear labelling and the imaging do not interfere with NF-κB signalling dynamics, so we expected the same for NF-κB-driven transcription. To further confirm this, we compared RT-PCR quantifications of transcription in cells stimulated with TNF and cultured in imaging medium (0.1% FCS+Hoechst) or standard MEF medium (10% FCS). The results for nascent and mature IκBα and Ccl5 transcripts reported in *Figure 5—figure supplement 2* show no difference in the transcriptional response in the two conditions.

## Microarray experiments

RNA samples were extracted using the IllustraRNAspin Mini kit (GE Healthcare). Following extraction, RIN (RNA Integrity Number) was >9 (BioAnalyser, Agilent RNA Nano Kit). RNA samples (500 ng) were reverse transcribed with the IlluminaTotalPrep RNA Amplification Kit (Ambion) and copy RNA (cRNA) was generated with 14 hr in vitro transcription reactions and checked at the BioAnalyser. Washing, staining, and hybridization were performed according to standard Illumina protocols. cRNA samples were then hybridized to IlluminaBeadChip Array MouseRefSeq-8 v2. BeadChips were scanned with BeadArray™ Reader in channel 2. The data have been uploaded on the Dryad Digital Repository (*Zambrano et al., 2016*). Experiments were repeated twice.

## Bioinformatic analysis of microarray experiments

Genome Studio's bead summary probe level data – not normalized and not background corrected – were analyzed using Bioconductor. We performed **quality assessment** by plotting the intensities of regular and control probes. **Sample intensities** were quantile normalized and filtered for expression and probe quality using beadarray R package: only probes with detection P-value <0.05 in at least one condition and whose categories is defined "Perfect" or "Good" were kept. Probes were labelled as deregulated if their absolute log2 fold change relative to the 0 time point were greater than 1.

**Clustering** was performed applying a soft clustering of deregulated probes with the Mfuzz R package, which is suggested for microarray time course-data (*Kumar and Futschik, 2007*). The optimal number of clusters was assessed with the d.min function. To focus on the **shape of the gene expression profiles**, we standardised the gene expression profiles by taking the usual base-two logarithm and normalizing to obtain mean 0 and standard deviation 1. The algorithm then groups genes based on the Euclidean distance between profiles and the c-means objective function, which is a weighted square error function. Each gene is assigned a membership value between 0 and 1 for each cluster. Hence, genes can be assigned to different clusters in a gradual manner. The parameters m defines the degree of "fuzzification". It is defined for real values greater than 1 and the bigger it is the more fuzzy the membership values of the cluster. We used an m value estimated by the "mestimate" function of 1.5. **Membership values** indicate the similarity of vectors to each other

defining a cluster cores. To extract list of genes belonging to the cluster cores, we used the "acore" function taking from each clusters all genes with a membership value of at least 0.5.

*Pathway analysis* of genes contained in each cluster was performed with the ClusterProfiling R packages using an adjusted p-value cut-off for enrichment <0.2 based on the hypergeometric distribution (*Yu, 2015*). We used for our analyses the Reactome database. In order to simplify the resulting output we plotted only the top five categories of the second and third level (*Yu, 2015*). Conversely, overlaps were calculated based on all categories of the second and third levels. The overlap coefficient (or Szymkiewicz-Simpson coefficient) between gene sets X and Y is given by:

$$overlap\,(X,Y) = \frac{|X \bigcap Y|}{\min(|X|,\,|Y|)}$$

For *statistical significance* we performed a Fisher's Exact Test for evaluating if the resulting clusters where enriched for NF-κB's targets. The resulting p-value has been converted in a significance measure (-Log$_{10}$(p value)). We used both a list taken from *Li et al (2014)* and from Thomas Gilmore's website, Boston University (http://www.bu.edu/nf-kb/gene-resources/target-genes/).

## Mathematical modelling

We propose here a mathematical model based on the one discussed in *Zambrano et al (2014b)* that adds the layer of regulation by considering the negative feedback provided by the protein A20 that blocks IKK activation upon stimulation (*Figure 1—figure supplement 4*). For the sake of completeness, we describe briefly below the basic process that we considered, together with the biochemical rates involved (values are given in *Supplementary file 2*) and a summary of our normalization procedure. Additional details on the motivations underlying certain selection of biochemical reactions and variables of interest can be found in that paper. Being a simple model, it provides important qualitative insights on the possible variety of single-cell dynamics, but we also use it to provide quantitative fittings of the average population dynamics and transcription.

## NF-κB activation and IκBα feedback

The biochemical reactions described here are essentially the basic ones given in the simple model given in *Zambrano et al (2014b)*. The basic simplification of our model is to consider that free NF-κB is nuclear, while the complex with the inhibitor IκBα is cytoplasmic. The formation of the complex is given by (*NF-κB:IκBα*)

$$NF - \kappa B + I\kappa B\alpha \xrightarrow{A} (NF - \kappa B : I\kappa B\alpha)$$

that can also spontaneously dissociate

$$(NF - \kappa B : I\kappa B\alpha) \xrightarrow{d} NF - \kappa B + I\kappa B\alpha$$

An external signal can lead to the appeareance of *IKK*$_a$ that can free *NF-κB* by degrading of *IκBα* in the complex

$$(NF - \kappa B : I\kappa B\alpha) + IKK_a \xrightarrow{P} NF - \kappa B + IKK_a$$

and in its free form

$$I\kappa B\alpha + IKK_a \xrightarrow{\kappa P} IKK_a.$$

The negative feedback loop is enabled by the fact that the gene encoding *IκBα*, $G_\alpha$, can be activated by *NF-κB*

$$G_{\alpha,OFF} + NF - \kappa B \xrightarrow{K_{on,I}} G_{\alpha,ON} + NF - \kappa B$$

while the inactivation is modulated by

$$G_{\alpha,ON} + I\kappa B\alpha \xrightarrow{K_{off,I}} G_{\alpha,OFF} + I\kappa B\alpha.$$

Notice that we are assuming here that *IκBα* plays a role as transcriptional repressor, as suggested by experimental results (*Arenzana-Seisdedos et al., 1995*) and used in some mathematical models

(*Kellogg and Tay, 2015*; *Lipniacki et al., 2007*; *Tay et al., 2010*), but not others (*Ashall et al., 2009*; *Sung et al., 2014*). We show later that this does not have a strong impact on the fittings, but to facilitate the correspondence with our previous work and in line with the mentioned modelling approach, we opt to keep this process for all the gene inactivations considered in most of our explorations. We only briefly explore the effect of setting the $k_{off}=0$ in *Figure 6—figure supplement 7*, results are detailed in the text.

We also consider that the gene can be basally activated

$$G_{\alpha,OFF} \xrightarrow{k_{on0,I}} G_{\alpha,ON}$$

and inactivated

$$G_{\alpha,ON} \xrightarrow{k_{off0,I}} G_{\alpha,OFF}$$

Transcription (mRNA production) is given by

$$G_{\alpha,ON} \xrightarrow{K_{RI}} G_{\alpha,ON} + I\kappa B\alpha_{RNA}$$

while the RNA degradation is given by

$$I\kappa B\alpha_{RNA} \xrightarrow{d_{RI}} \emptyset.$$

$I\kappa B\alpha$ translation is given by

$$I\kappa B\alpha_{RNA} \xrightarrow{K_I} I\kappa B\alpha$$

while its spontaneous degradation is given by

$$I\kappa B\alpha \xrightarrow{d_I} \emptyset,$$

a degradation that is also possible while it is forming the complex

$$(NF-\kappa B : I\kappa B\alpha) \xrightarrow{\gamma d_I} NF-\kappa B$$

## IKK activation and A20 feedback

The protein *A20* is known to provide a negative feedback by modulating *IKK* activation process (*Ashall et al., 2009*). We summarize this in our new mathematical model using the following processes in which we just model the evolution of the active form, $IKK_a$.

Given a external time dependent *TNF-α* variation, *TNF(t)*, characterized by certain values of the alternating doses, $D_1$ and $D_2$ in times $T_1$ and $T_2$, we will assume that the appearance of the active IKK, $IKK_a$, can be summarized as:

$$TNF(t) \xrightarrow{K(A20)} IKK_a.$$

In a situation of constant flow of TNF-α the value of *TNF(t)* would be constant and equal to a dose *D*.

The biochemical rate of this equation, *K(A20)*, is the only rate that we consider as variable. In doing so, we aim to summarize in just one biochemical reaction the fact that *A20* contributes to block *IKK* activation (*Ashall et al., 2009*) instead of modelling the whole *IKK* activation module, so

$$K(A20) = \frac{K_S}{1 + \left(\frac{A20}{A20_0}\right)^n}$$

*IKK* gets spontaneously inactivated as

$$IKK_a \xrightarrow{d_K} \emptyset.$$

The gene encoding *A20* is regulated by the same activation and inactivation processes as *IκBα* so

$$G_{A20OFF} + NF-\kappa B \xrightarrow{K_{on,A20}} G_{A20,ON} + NF-\kappa B$$

$$G_{A20,ON} + I\kappa B\alpha \xrightarrow{K_{off,A20}} G_{A20,OFF} + I\kappa B\alpha$$

$$G_{A20,OFF} \xrightarrow{K_{on0,A20}} G_{A20,ON}$$

$$G_{A20,ON} \xrightarrow{K_{off0,A20}} G_{A20,OFF}$$

Transcription is given by

$$G_{A20,ON} \xrightarrow{K_{R,A20}} G_{A2,ON} + A20_{RNA}$$

while the RNA degradation is given by

$$A20_{RNA} \xrightarrow{d_{R,A}} \emptyset$$

finally *A20* is translated as

$$A20_{RNA} \xrightarrow{K_A} A20$$

and it is spontaneously degraded as

$$A20 \xrightarrow{d_A} \emptyset$$

## Transcription of an NF-κB controlled gene

As for the genes encoding for *A20* and *IκBα*, we consider that for any *NF-κB* controlled gene transcription is regulated by the processes

$$G_{OFF} + NF-\kappa B \xrightarrow{K_{on,G}} G_{ON} + NF-\kappa B$$

$$G_{ON} + I\kappa B\alpha \xrightarrow{K_{off,G}} G_{OFF} + I\kappa B\alpha$$

$$G_{OFF} \xrightarrow{K_{on0,G}} G_{ON}$$

$$G_{ON} \xrightarrow{K_{off0,G}} G_{OFF}$$

while the RNA produced from the gene is provided by

$$G_{ON} \xrightarrow{K_R} G_{ON} + A20_{RNA}$$

and its degradation is given by

$$G_{RNA} \xrightarrow{d_{R,G}} \emptyset$$

## Model equations, normalization and parameter selection and uncertainty

The equations of the model are derived using mass-action kinetics and performing a normalization in much the same way as in *Zambrano et al (2014b)*. We describe below the process, using the same symbols to represent biochemical species and their copy number.

First, as in previous existing models and as suggested by our experiments, the total amount of NF-κB, free plus bound to the inhibitor, remains constant and equal to NF-κB$_0$ so we define $N=NF-\kappa B/NF-\kappa B_0$. We also normalize the amount of inhibitor as $I= NF-\kappa B/NF-\kappa B_0$ and the amount of *A20* as $A= A20/A20_0$ and $K=IKK/IKK_0$, where $A20_0$ and $IKK_0$ are amounts of reference in such a way that $N$, $I$ and $A$ are adimensional variables of the order of 1. On the other hand, for any gene we have that the maximum number of on states is $G_0=2$ (the two alleles), so $G_I$, and $G_A$ are a number between 0 and 1 representing the fraction of active genes encoding for *IκBα* and for *A20*, respectively – it is a continuous approximation to a discrete variable. Finally, notice that using our notation the maximum number of active genes is $G_0$ so it can be proved that the maximum asymptotic value of the amount of RNA of the biochemical species $X$ is of the form $K_{R,X} \cdot G_0/d_{R,X}$. Hence, we define the

variables $R_I$, and $R_A$ as proportional to the amount of mature RNA of $I\kappa B\alpha$ and $A20$, respectively, via the relations

$$R_I = I\kappa B\alpha_{RNA} \cdot d_{R,I}/(K_{RI} \cdot G_0) \text{ and } R_A = A20_{RNA} \cdot d_{R,A20}/(K_{R,A20} \cdot G_0).$$

Overall, these normalizations allow one to pass from biochemical species with copy numbers of different orders of magnitudes to adimensionalized variables that are of the order of 1.

By using them, and renaming combinations of constants (in such a way that lowercase parameters are obtained from uppercase biochemical reaction rates), we have the following model of ordinary differential equations describing the dynamics of our system:

$$\frac{dK}{dt} = -d_K \cdot K + \frac{1}{1+(A)^n}S(t) \tag{1}$$

$$\frac{dN}{dt} = d \cdot (1-N) + \gamma \cdot d_I(1-N) + p \cdot K \cdot (1-N) - a \cdot I \cdot N \tag{2}$$

$$\frac{dG_I}{dt} = \left(k_{onI}N + k_{on,0,I}\right)(1-G_I) - \left(k_{offI}I + k_{off,0I}\right)G_I \tag{3}$$

$$\frac{dR_I}{dt} = d_{RI}(G_I - R_I) \tag{4}$$

$$\frac{dI}{dt} = d \cdot (1-N) - \kappa \cdot p \cdot K \cdot I - a \cdot I \cdot N + k_I \cdot kR_I - d_I \cdot I \tag{5}$$

$$\frac{dG_A}{dt} = \left(k_{on,A}N + k_{on,0,A}\right)(1-G_A) - \left(k_{off,A}I + k_{off,0,A}\right)G_A \tag{6}$$

$$\frac{dR_A}{dt} = d_{RA}(G_A - R_A) \tag{7}$$

$$\frac{dA}{dt} = k_A \cdot R_A - d_A \cdot A \tag{8}$$

The parameters used as starting point in our explorations and the fittings can be found in the source code folder for mathematical modelling, while the original biochemical rates from which they were derived can be found in *Supplementary file 2*. Normalised parameter values are provided in the source codes. Many of them have the same values as the ones provided in *Zambrano et al (2014b)*. Additional parameters were manually fitted or extracted from (*Tay et al., 2010*), as we did previously. Notice that the degradation rates of the kinase, the RNAs and the proteins appear as parameters in the equations above, unchanged. The remaining parameters are related to the original biochemical rates as $p=P \cdot IKK_0$, $a=A \cdot NF\text{-}\kappa B_0$, $k_I=K_I \cdot K_{R,I} \cdot G_0/(d_{R,I} \cdot NF\text{-}\kappa B_0)$, $k_A=K_A \cdot K_{R,A} \cdot G_0/(d_{R,A} \cdot A20_0)$, $k_{on,I}=K_{on,I} \cdot NF\text{-}\kappa B_0$, $k_{off,I}=K_{off,I} \cdot NF\text{-}\kappa B_0$, $k_{on,A}=K_{on,A} \cdot NF\text{-}\kappa B_0$, $k_{off,A}=K_{off,A} \cdot NF\text{-}\kappa B_0$. $S(t)$ is the normalized signal, in such a way that a constant TNF = 10 ng/ml is equivalent to $S(t)=2 \ h^{-1}$ in the system of equations. For lower doses we use lower values of $S$.

Notice that the adimensionalization leads to a number of parameters smaller than the total number of biochemical rates and constants of the original system. *Equations 2–5* are identical – except for the inclusion of spontaneous gene activation-inactivation – to the ones proposed in our previous minimal model of the regulatory network (*Zambrano et al., 2014b*). *Equation 1* models the effect of A20 as inhibitor of the kinase activation. This second negative feedback is completed with *Equations 6–8*, that explicit NF-κB control of A20 expression.

As shown below, we use this model both to fit the observed dynamics and for numerical exploration of the dynamics observed under different stimuli. For simplicity, we call $P_{NF\text{-}\kappa B}$ to the set of parameters of *Equations 1–8* and $P_S$ to the parameters describing the parameters of the external signal, that can either be constant or a time varying rectangular forcing signal as shown in *Figure 1A*. Following what we did in previous numerical explorations (*Zambrano et al., 2014b*) we associate to each parameter an *uncertainty degree D*, so each parameter can be randomly varied by multiplying it by a factor in the interval $[10^{-D}, 10^D]$ with D smaller or equal than 1. In other words, parameters with higher uncertainty degree can be varied up to one order of magnitude above or below their selected initial value, which is itself a moderate variation. Those degrees are specified in *Supplementary file 2*: note that we choose higher values of D for manually fitted parameters and

lower for those from the literature or from our own measurements. We also consider a high value of $D$ for parameters that summarize many different processes or for those in which our model is less detailed, as in the IKK activation – A20 regulatory module.

Finally, the equations for a gene under the control of *NF-κB* are:

$$\frac{dG}{dt} = \left(k_{on,G}N + k_{on,0,G}\right)(1 - G) - \left(k_{off,G}I + k_{off,0,G}\right)G \tag{9}$$

$$\frac{dR}{dt} = d_{R,G}(G - R) \tag{10}$$

where in this case $R = G_{RNA} \cdot d_{R,G}/(K_R \cdot G_0)$, $k_{on,G} = K_{on,G} \cdot NF\text{-}\kappa B_0$, $k_{off,G} = K_{off,G} \cdot NF\text{-}\kappa B_0$. The parameters in (9) and (10) are used to model and fit different expression profiles are denoted as $P_G$. Notice that to model the process of gene expression one needs to set $P_S$, $P_{NF\text{-}\kappa B}$ and $P_G$. In this context, two different genes expressed under the same external stimuli conditions would share the parameters $P_S$ and $P_{NF\text{-}\kappa B}$ but different values of $P_G$. This idea is used in our fittings of dynamics and transcription in this work.

## Classification of the different dynamical responses for constant stimulus

We have observed experimentally that even for constant stimulation different dynamics are possible, with trajectories showing different degrees of dampening and sometimes even irregular and non-oscillating profiles (*Figure 1—figure supplement 2D*) as pointed out in previous works (*Sung et al., 2009*; *Zambrano et al., 2014b*). Our relatively simple model illustrates why this situation might arise and how a wider variety of dynamics might arise compared to the regular oscillations documented in the literature for more complicated models (e.g. (*Ashall et al., 2009*; *Tay et al., 2010*)).

To illustrate this, we analyzed the dynamics for a constant external stimulus ($S(t)=2$ h$^{-1}$) and varying simultaneously each parameter of the parameter set $P_{NF\text{-}\kappa B}$ according to their uncertainty degree D. We used them to generate trajectories and selected those giving rise to a response, i.e. those for which N(t) is bigger than 0.4 in the first 3 hr and whose average values in the last hours was smaller or equal than N=0.4. To characterized in a systematic way their dynamics, we located the fixed point of *Equations 1–8* and calculated the eigenvalues {$\lambda_i$} (i=1,...8) of the Jacobian. In *Figure 1—figure supplement 5A* we plot the real and imaginary parts of each set of eigenvalues. We represent with a red dot the eigenvalues belonging to a set in which the real part of at least one of them is bigger than zero. This means that the fixed point for those parameters is unstable and hence trajectories converge to a stable limit cycle around it. In other words, for those parameter combinations sustained oscillations arise. The percentage of parameter sets giving sustained oscillations is below 10% (*Figure 1—figure supplement 5B*, right panel), which make us conjecture that parameters giving rise to sustained oscillations are not the norm.

We also plot in *Figure 1—figure supplement 5C* the parameter values giving rise to oscillating and non oscillating (but responding) trajectories. We note that the intervals are very similar except for parameters n and $d_A$ involved in the A20 negative feedback, which indicates that this feedback is critical in order to obtain a sustained or a damped oscillatory response. In previous works it was shown experimentally and through mathematical models that the interplay between the IκB and A20 regulatory modules regulate different phases of NF-κB response (*Werner et al., 2008*). Our numerical results suggest that it also plays a key role in the type of dynamics observed (oscillatory versus damped). Overall, this analysis further hints to the fact that it is the precise combination of parameters, rather than the precise single values, what determines the type of dynamics observed and might account for the different dynamics observed with respect to other groups (*Ashall et al., 2009*; *Cheng et al., 2015*; *Kellogg and Tay, 2015*; *Lee et al., 2009*; *Sung et al., 2009*; *Tay et al., 2010*).

Finally, we have to notice that being the system given by *Equations 1–8* a dynamical system with real variables, the imaginary eigenvalues come in complex conjugate pairs. The distribution of parameter combinations with 2, 4 and 6 imaginary eigenvalues are shown in *Figure 1—figure supplement 5B* (left panel). Notice that an oscillatory frequency can be associated to each couple of complex conjugate pairs (*Goldstein et al., 2001*), so having more than one of such couples means that the dynamics will combine different frequencies.

To further illustrate all this, in *Figure 1—figure supplement 5D* we present examples of the variety of trajectories found. Some of them are oscillating (red), showing both smooth and spiky

oscillatory peaks. Many others are damped and in some cases the concurrence of two oscillating timescales, fast and slow, is evident. Some others present clearly non-oscillating profiles. Overall, our numerical exploration of our relatively simple models shows that different dynamics are possible in presence of constant stimulation, as observed in the experiments of this and previous works (*Sung et al., 2009*; *Zambrano et al., 2014a*).

The model simulation routine was implemented in C code language and compiled using gcc. Routines for parameter variation and time series analysis and stability analysis (using the function fsolve of the Nonlinear Equations package) where written using GNU Octave.

## Parameter fitting

Routines were written in GNU octave to fit the model to the data by combining Markov Chain Monte Carlo for initial exploration of the parameter space and a Levenberg–Marquardt algorithm. Parameters of *Equations 1–8* were varied within the specified uncertainty degree. The goal is to minimize the distance between a given goal signal $X=\{X(t_k)\}$ for $k=1,...,N$ and the theoretical values $x=\{x(t_k)\}$ obtained from the model. We define the distance of a given fitting to the data as:

$$d_{X,x} = \frac{1}{N} \sum_k \frac{|X(t_k) - x(t_k)|}{\max\left(X(t_k), x(t_k)\right)}$$

When several data are fitted simultaneously, the total distance between model predictions and the data were obtained by summing each distance. Notice that the $1/N$ factor weighs for difference in the length of the data. For a given computations, this distance gives the average relative error per timepoint of the fitting.

- To fit NCI, theoretical NCI profiles from the model were obtained as:

$$NCI(t) = \frac{S_{cyto}}{S_{nucleus}} \frac{N(t)}{1 - N(t)}.$$

where $S_{cyto}/S_{nuc}$ is the ratio of the nuclear to total area of the cells in our image, which is estimated to be of around 1/3.

For parameter fitting of the gene expression obtained from Quantitative Real-Time and microarray experiments (see below), the fold change levels predicted by the models where computed as $R(t)/R(0)$. For the simultaneous fitting of NCI, IκBα and Cccl5 mRNA shown in *Figure 5*, common $P_{NF-κB}$ parameters where found while for IκBα and CCL5 mRNA we independently fitted the respective set of parameters $P_G$. For the fitting of $P_{NF-κB}$ the parameters were varied within the limits given by their uncertainty degree.

Fitting of the microarray data was performed by keeping constant the parameters $P_{NF-κB}$ obtained fitting the data of *Figure 5* and varying $P_G$ (those of (9–10)) for each gene, as we did for IκBα and Cccl5. Examples are shown in *Figure 6—figure supplement 6*. *Figure 6—figure supplement 5* shows that the distance between fitting and real data is better for genes with increased transcription with an average fitting error of 9%.

## Acknowledgements

SZ was supported by the Intra-European Fellowships for career development-2011–298447Non-LinKB. The work was supported by the IEF-2011-298447 research funding to SZ, the Italian AIRC 2010/2013 project R0444 and Flagship Program EPIGEN to MEB. We are grateful to Rohini Nair and Lucianna Calia for help in the experiments reported in *Figure 1—figure supplement 11* and in *Figure 5—figure supplement 2*, respectively. We are in debt with Nacho Molina and Davide Mazza for suggestions and critical reading of the manuscript. We also thank the Millipore service for assistance in microfluidics set-up. We thank the reviewing team for their help and constructive comments.

## Additional information

### Funding

| Funder | Grant reference number | Author |
|---|---|---|
| European Marie Curie Action 2011 | 298447-NonLinKB | Samuel Zambrano |
| Intra European Fellowships | MCA-2011-298447NonLinKB | Samuel Zambrano |
| AIRC2010/2013 project R0444 | | Marco E Bianchi |
| Flagship program EPIGEN | | Marco E Bianchi |

The funders had no role in study design, data collection and interpretation, or the decision to submit the work for publication.

### Author contributions

SZ, Performed the experiments, Mathematical modelling and model fitting, Analysis of the dynamics, Conception and design, Acquisition of data, Analysis and interpretation of data, Drafting or revising the article; IDT, Performed the experiments, Analysis of the dynamics, Bioinformatics analysis, Acquisition of data, Analysis and interpretation of data, Drafting or revising the article; AP, Performed the experiments, Mathematical modelling and model fitting, Analysis of the dynamics, Analysis and interpretation of data; MEB, Analysis of the dynamics, Conception and design, Analysis and interpretation of data, Drafting or revising the article; AA, Performed the experiments, Analysis of the dynamics, Bioinformatics analysis, Conception and design, Acquisition of data, Analysis and interpretation of data, Drafting or revising the article

### Author ORCIDs

Marco E Bianchi, http://orcid.org/0000-0002-5329-6445
Alessandra Agresti, http://orcid.org/0000-0001-5006-9506

## Additional files

### Supplementary files

• Supplementary file 1. RT-PCR primers list. Listed are the primers used in Q-PCR reaction to test gene expression as reported in *Figure 5* and in *Figure 5—figure supplement 2* and *Figure 7—figure supplement 1*.

• Supplementary file 2. Biochemical rates for the model. In this file we indicate the biochemical rates and the constants considered, which are described in the Materials and methods. We also provide the value of these rates and the bibliographic reference from which they were taken, if any. Otherwise, we motivate our selection of the value or we state that they were manually fitted. We also indicate the related model parameters resulting from the adimensionalization of the equations of the dynamics and the uncertainty degree, a measure of how uncertain each parameter value is. This is also a measure of how much we allow each parameter to vary in our exploration of the dynamics of the system and in our fittings. Notice that higher uncertainty degrees are assigned to parameters that were manually fitted or to those that are involved in reactions from our models that summarize many different biochemical processes.

### Major datasets

The following dataset was generated:

| Author(s) | Year | Dataset title | Dataset URL | Database, license, and accessibility information |
|---|---|---|---|---|
| Zambrano S, De Toma I, Piffer A, Bianchi ME, Agresti A | 2016 | Data from: NF-κB oscillations translate into functionally related patterns of gene expression | http://dx.doi.org/10.5061/dryad.j62n1 | Available at Dryad Digital Repository under a CC0 Public Domain Dedication |

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
