## [Decision Letter]

Thank you for submitting your work entitled "NF-κB oscillations translate into functionally related patterns of gene expression" for peer review at *eLife*. Your submission has been favorably evaluated by Aviv Regev (Senior editor), a Reviewing editor, and two reviewers.

The reviewers have discussed the reviews with one another and the Reviewing editor has drafted this decision to help you prepare a revised submission

Summary:

How transcription factors affect responses to extracellular signals remains a key question in systems biology for which the NF*-*κB signaling pathway in mammalian cells has been a canonical study system. The manuscript "NF-κB oscillations translate into functionally related patterns…" by Zambrano et al. presents a systematic investigation of single-cell NF-κB signaling dynamics and of population-level transcriptional responses in GFP-RelA knock-in mouse fibroblasts stimulated with controlled TNF pulses. The authors use a microfluidic device to deliver TNF to cells while avoiding the accumulation of secreted proteins. Although, this does not exclude cell-centric feedback regulation of the signaling response to TNF, the minimal ODE model proposed by Zambrano et al. nonetheless accounts well for their experimental observations and the authors conclude that NF-κB behaves as a damped oscillator in these fibroblasts. That said, the reviewers have raised serious concerns about the single-cell data (detailed below) that need to be addressed.

The second half of the study, focused on the gene expression kinetics and functional correlates in cells treated with TNF pulses, leverages the use of oscillatory stimulations to reveal different clusters of genes that share dynamic patterns of expression. This is the most convincing and interesting finding of the manuscript. An important biological insight is that different functional pathways exhibit different expression dynamics; chemokine/chemokine receptors oscillate with TNF inputs, while immune responses integrate over longer timescales. The authors convincingly show that the decay rate of the mRNA of a gene of interest is sufficient to account for the different dynamic regimes of gene regulation.

Essential revisions:

1) The live cell imaging conditions seem to have been very damaging to cell physiology. Stress from photo-damage during live-cell imaging is a very serious issue which has hampered many cell biological studies (Cole R, 2014 Cell Adhesion & Migration 8 (2) 452-259), and it is essential to address this problem. In the data presented by Zambrano et al., the supplementary movies show that the knock-in fibroblasts express very low levels of GFP-RelA making imaging challenging and it is quite evident from these movies that there is significant cell death and stress, and few cell divisions. Photo-stress was likely increased in this study due to the repeated exposure to the UV laser used for Hoechst imaging. A key concern is that with such a high level of stress, the observed NF-κB dynamics may not represent the true signaling dynamics induced solely by TNF. This could explain the authors observe highly damped oscillations, which are different from the sustained oscillations observed and reported in their first paper on the same GFP-RelA knock-in MEFs (Sung MH et al. 2009 PLoS One). In contrast to the manuscript under consideration, no UV or Hoechst were used in the 2009 study and healthy cell divisions were frequently observed throughout the long-term imaging supplementary movies. In light of these important concerns, the inclusion of the single-cell imaging data in the paper requires additional control data:

A) Because nuclear dyes can alter cell cycle and/or cause oxidative stress and mutations (Martin RM, Leonhardt H, and Cardoso MC, 2005 Cytometry; Ge J, Wood DK, et al. 2013 Cytometry), crucial control experiments should be performed to tease out the effects of the long incubation with the dye and the effects of UV radiation, with or without TNF. Do the dye and UV light perturb natural signaling dynamics or other physiology of the cells, such as cell cycle and apoptosis?

B) Importantly, the authors should compare the transcriptional response of cells with or without exposure to dye and UV light to verify that similar results are obtained under both conditions.

2) With regards to the quantification of nuclear GFP-RelA, the authors state that the nuclear to cytoplasmic intensity, termed NCI, is an "internally normalized measure". Although this quantity has been used previously to represent nuclear localization of NF-κB (Nelson DE et al. Science 2004 and subsequent publications from the same group), more groups now approximate the nuclear concentration of NF-κB with the background-adjusted mean nuclear intensity (Lee TK et al. 2009 Science Signaling; Sung MH et al. 2014 Science Signaling; Lee REC, et al. 2014 Molecular Cell; Kellogg RA and Tay S, 2015 Cell), while minimizing the photobleaching which decreases fluorescent signals. NCI is not fully internally normalized, because both the numerator and the denominator fluctuate over time due to biological effects, not only due to technical effects. Would the same damped oscillations dynamics be observed with background-adjusted mean nuclear intensity?

3) Regarding the comparison with the recent report by Kellogg and Tay (2015 Cell), one important difference that is not highlighted here is that the applied pulse characteristics are different. The Tay group delivered a pulse by an infusion of TNF-α without washing (i.e. T2 = 0, using the terminology of Figure 1), assuming a clearance process through internalization and degradation. Zambrano et al. enforced a washing and reset period between consecutive pulses. While it is a matter of debate which method better mimics a physiologically relevant situation, the different forcing protocols may have affected their results and make a direct comparison problematic. Could this explain why the two groups observe different results?

4) Concerning the choice of pulse duration for different forcing frequencies, the authors should justify why different "T1" periods were used for different forcing frequencies (45 min for 90 min pulses, 30 min for 180 min pulses, etc.). Michael White's group has reported that the temporal profile of nuclear NF-κB depends on the pulse duration (Ashall L et al. 2009 Science, FigureS7). Fixing the pulse duration might simplify the interpretation of results from a systematic comparison.

5) With regards to the computational model, one concern is that many parameter values seem ill-defined experimentally. Although the model builds on a previously published version, a more exhaustive presentation of the model (to explain which parameters are fixed, and which are fitted) should be included. It is also recommended that experimental effort be made to narrow down the number of free parameters: this greatly improves the quality of biochemical models, and makes for better predictions. Finally, IκBα is encoded in the model to exert a "transcriptional repressor" role although the cited study does not present particularly convincing evidence for any direct repressor activity of IκBα. Is this term necessary to recapitulate the experimental data, especially for a minimal mathematical model?

6) With respect to the sensitivity of model behavior to parameter values, the authors acknowledge large parameter sensitivity of their model, with 2-fold changes in expression levels of proteins switching cells from damped to sustained oscillatory responses (c.f. Figure 1—figure supplement 5).

A) An exact description of the parameters that were varied in these supplementary figures needs to be included.

B) More generally, a systematic parameter sensitivity of the model (e.g. in terms of generating varied types of oscillation) should be performed to allow readers to understand the robustness of the model (in terms of its predictive power). In particular, this could potentially identify key proteins whose up/down regulation would most affect the response phenotype. In turn, the natural variability in the expression levels of these signaling components should be taken into account: it would explain the phenotypic variability that must exist within the isogenic population of cells used in these experiments, but that was overlooked so far.

C) Most relevant to the biology of NFκB signaling, such parameter sensitivity would help reconcile the diverse dynamics of NFκB response depending on the cell type under consideration (c.f. comparison with Kellogg and Tay, 2015 study). For that reason, it would be particularly useful to document systematically the variety of phenotypes (from damped oscillations, to spontaneous oscillations etc.) to predict and explain how different cell types use NFκB response differently to generate different functional responses, and to support the authors' argument that damped oscillations responses may be common while the stable oscillations may only be a rare case.

[Editors' note: further revisions were requested prior to acceptance, as described below.]

Thank you for resubmitting your work entitled "NF-κB oscillations translate into functionally related patterns of gene expression" for further consideration at *eLife*. Your revised article has been favorably evaluated by Aviv Regev (Senior editor) and a Guest Reviewing editor. The manuscript has been improved but there are some remaining minor issues that need to be addressed before acceptance, as outlined below.

In their revised submission, Zambrano and colleagues have addressed most of the concerns raised in the original review of this manuscript and the submission is accordingly much stronger. In particular, the authors have added a number of datasets to validate their experimental setup for live-cell imaging, mitigating initial concerns about damaging conditions for the cells under study. The new justification in the main text (Results) along with new supporting data (Figure 1 supplements) better explains the rationale for using this particular measure of nuclear NF-κB in this study. Furthermore, the detailed explanation now included in the Materials and methods presents a complete argumentation for this approach with their current datasetMaterials and methods, and clearly lays out the assumptions made and therefore provides an excellent overview for all readers familiar, or unfamiliar, with these types of data. Additional data and explanations in the text also better support the choice of stimulation profile. The new manuscript also provides a well-done analysis of their computational model which helps position their findings with regards to the dynamic behavior of TNF-induced NF-κB responses in a greater context of possible and probably behaviors. Only a few minor issues are noted for the revised version which we hope can be rapidly addressed by the authors.

Cell death, cell division and imaging conditions – overall, the additional data provided by the authors, particularly Figure 1—figure supplement 7 and Figure 5—figure supplement 2 do suggest that the cell death that is observed in some of the experiments is likely due to the presence of TNF (under flow condition and in the absence of serum), and not just an artefact of the use of a Hoescht dye and UV illumination to identify the nucleus of each cell. However, the evidence from PO4-gammaH2AX in Figure 1—figure supplement 6 has an important weakness and should probably be de-emphasized. Indeed, this epitope marks double strand breaks (DSBs) and while doxorubicin (positive control) is known to produce DNA double strand breaks (DSB) in cells in most phases of the cell cycle (DSB are linked to genes actively transcribed; e.g. http://www.ncbi.nlm.nih.gov/pubmed/25705119) UV illumination creates other types of DNA damage (pyrimidine dimers) but only lead to DSBs in cells in S-phase and on a longer time scale than that used bythe authors in this figure (6-12 hrs, vs. 3 hrs – e.g. http://www.ncbi.nlm.nih.gov/pubmed/20947453 and http://www.ncbi.nlm.nih.gov/pubmed/18800352). Therefore, lack of PO4-gammaH2AX staining does not necessarily preclude the possibility of UV-induced DNA damage.

Square vs. sawtooth stimulation profile. Because the geometry of the microfluidic chamber is very different in this study from that of the device used in Kellogg and Tay, it is unclear if a significant reduction in TNF concentration is achieved when medium flow is stopped (as noted by the authors, TNF is stable in their chamber, and would decay only if actively degraded by cells). Importantly, this could be an alternative explanation for the lack of synchrony – this stimulation profile may be closer to static concentration than to oscillatory stimulation – which should be noted in the discussion of these data.

Regarding the sensitivity analysis of the model – the sensitivity analysis does provide a greater context for the observations by the authors as well as the cited work by Kellogg and Tay. The authors mention that parameters related to the A20 feedback are the only one for which there is a clear difference between values that lead to different types of dynamics. When discussing this result, the authors may find it helpful to refer to Werner et al. (2008; Genes and Dev.), where the A20 feedback was explored computationally and experimentally. This paper should also be cited in the fourth paragraph of the Introduction, when introducing the A20 feedback.

Regarding the role of IκBα as a repressor – the new results from the authors do shed more light on that issue. However, the authors should review their model diagram as some are ambiguous or possibly misleading. It would be easier for the readers to interpret the model if the "activation" arrow for NF-κB and the "inhibition" arrow for IκBα were carefully positioned to target the reactions on which they act.

In the third paragraph of the subsection “Periodic forcing turns heterogeneous NF-κB oscillations into synchronous oscillations”:

a) "Replicate and show…"; "divide" or "proliferate" would be a better term to characterize the cells' behavior.

b) “Limited cell death… "; it may be more appropriate to just state "cell death…".

In the fourth paragraph of the Introduction: "In resting cells, p65:p50 exists mostly as a cytoplasmic complex bound to the IκB inhibitors (Hoffmann et al., 2002)" would be better rephrased to: "In resting cells, p65 exists mostly within a cytoplasmic complex bound to the IκB inhibitors (Hoffmann et al., 2002)."

"Figure 4: The authors should indicate in the main text or in the figure legend how many forcing cycles the cells were stimulated with before withdrawal of the forcing." The authors have now added cycle numbers on the figures although it would be helpful if the figure legend explained what these numbers represent.

---

## [Author Response]

Summary:

*How transcription factors affect responses to extracellular signals remains a key question in systems biology for which the NF-κB signaling pathway in mammalian cells has been a canonical study system. The manuscript "NF-κB oscillations translate into functionally related patterns*…*" by Zambrano et al. presents a systematic investigation of single-cell NF-κB signaling dynamics and of population-level transcriptional responses in GFP-RelA knock-in mouse fibroblasts stimulated with controlled TNF pulses. The authors use a microfluidic device to deliver TNF to cells while avoiding the accumulation of secreted proteins. Although, this does not exclude cell-centric feedback regulation of the signaling response to TNF, the minimal ODE model proposed by Zambrano et al. nonetheless accounts well for their experimental observations and the authors conclude that NF-κB behaves as a damped oscillator in these fibroblasts. That said, the reviewers have raised serious concerns about the single-cell data (detailed below) that need to be addressed. The second half of the study, focused on the gene expression kinetics and functional correlates in cells treated with TNF pulses, leverages the use of oscillatory stimulations to reveal different clusters of genes that share dynamic patterns of expression. This is the most convincing and interesting finding of the manuscript. An important biological insight is that different functional pathways exhibit different expression dynamics; chemokine/chemokine receptors oscillate with TNF inputs, while immune responses integrate over longer timescales. The authors convincingly show that the decay rate of the mRNA of a gene of interest is sufficient to account for the different dynamic regimes of gene regulation.* We thank the reviewers for the insightful comments, which highlight the need for important controls that were lacking or had not been mentioned in the previous version. We describe below how we have addressed the major concerns of the reviewers about the single-cell data and we describe the changes introduced in the manuscript.

Our controls reinforce the conclusion that NF–κB oscillations can lock to a wide variety of input stimuli as damped oscillations. We further argue that our isogenic cell population behave as heterogeneous damped oscillators for a constant stimulus, and this is supported by further imaging experiments and mathematical modelling.

*Essential revisions: 1) The live cell imaging conditions seem to have been very damaging to cell physiology. Stress from photo-damage during live-cell imaging is a very serious issue which has hampered many cell biological studies (Cole R, 2014 Cell Adhesion & Migration 8 (2) 452-259), and it is essential to address this problem. In the data presented by Zambrano et al., the supplementary movies show that the knock-in fibroblasts express very low levels of GFP-RelA making imaging challenging and it is quite evident from these movies that there is significant cell death and stress, and few cell divisions. Photo-stress was likely increased in this study due to the repeated exposure to the UV laser used for Hoechst imaging. A key concern is that with such a high level of stress, the observed NF-κB dynamics may not represent the true signaling dynamics induced solely by TNF. This could explain the authors observe highly damped oscillations, which are different from the sustained oscillations observed and reported in their first paper on the same GFP-RelA knock-in MEFs (Sung MH et al. 2009 PLoS One). In contrast to the manuscript under consideration, no UV or Hoechst were used in the 2009 study and healthy cell divisions were frequently observed throughout the long-term imaging supplementary movies. In light of these important concerns, the inclusion of the single-cell imaging data in the paper requires additional control data: A) Because nuclear dyes can alter cell cycle and/or cause oxidative stress and mutations (Martin RM, Leonhardt H, and Cardoso MC, 2005 Cytometry; Ge J, Wood DK, et al. 2013 Cytometry), crucial control experiments should be performed to tease out the effects of the long incubation with the dye and the effects of UV radiation, with or without TNF. Do the dye and UV light perturb natural signaling dynamics or other physiology of the cells, such as cell cycle and apoptosis?*

On one hand, these comments highlight the need to complete and make explicit the controls already made that exclude any interference of our nuclear labeling with the natural signaling of NF-κB in the cells. On the other hand, they suggest that our description of single cell dynamics as damped (and not sustained) oscillations was not as sharp as it should have been.

To address the first issue, we first performed several control experiments.

DNA damage:

To exclude DNA damage in the cells due to imaging, we assessed by immunostaining the presence of H2AX phosphorylation (gammaH2AX), a marker for ongoing DNA damage repair. We compared gammaH2AX staining in cells that were not exposed to Hoechst nor imaged and in those exposed to Hoechst and/or laser irradiation for imaging.

Figure 1—figure supplement 6 shows the results of this analysis: no difference in gammaH2AX intensity was detected in the cell nuclei upon different treatments, only the expected basal levels are observed (Turinetto & Giachino, 2015). It is also worth considering that NF-κB plays a pivotal role in activating the DNA damage response upon irradiation and a nuclear translocation would be expected if imaging is anyhow harmful. Importantly, we do not see increasing levels of global activation of NF-κB through the NCI time series for unstimulated cells (Figure 1—figure supplement 8 and Video 6).

Hence, at the low concentration of labeling used and the relatively low energy of our laser (405 nm) DNA damage remains almost undetectable. These results have been discussed in the main text and described in detail in the Materials and methods section.

Cell divisions:

It is important to point out that cells are cultured in medium containing reduced serum (0.1% FBS) to minimize background noise during imaging, a necessary technical condition when the signal to noise ratio is very low. In low serum, cells progressively transit through the G2/M phase and accumulate in G1, as documented by the cell cycle analyses reported in Figure 1—figure supplement 10. The low serum does not block the cell cycle completely and cells in mitosis are present until the very last imaging frame in all conditions tested. Cells are healthy as can be appreciated in the movies of unstimulated cells (Video 6). This is the main explanation for the low number of mitoses noted by the reviewers, whom we thank for asking for clarifications.

We acknowledge that a more accurate assessment of the imaging effect on the cell cycle would have been provided by single-cell based approaches like the FUCCI cell cycle sensor (https://www.thermofisher.com/it/en/home/life-science/cell-analysis/cell-viability-and-regulation/cell-cycle/live-cell-imaging-of-cell-cycle-and-division.html) that allows following cell divisions within a cell population during imaging. We actually tried this approach but our fibroblasts are very resistant to transfection. The minimal fraction of cells that were labeled with the sensor did not allow a statistically significant evaluation of cell cycle alteration, if any, in the cells undergoing imaging in the presence/absence of TNF-α.

Cell death:

In our previous work with these cells and imaging settings (Zambrano et al., 2014a) we quantified photo-stress after 15 hours of imaging of Hoechst-exposed (Video S1 and S2 in the paper) and non-exposed cells (Videos S3 and S4 of that paper), finding equally low death rates (close to 5%). Indeed, in microfluidics set-up, cell death is even lower in unstimulated cells than in Zambrano et al. 2014, see Video 6.

However, the reviewers are right in pointing out that Video 1 with cells treated with 10ng/ml TNF-α shows considerable cell death and now we propose an explanation for that effect. We suspected that TNF-α–but not Hoechst/UV imaging– might lead to cell death. To test this we evaluated the appearance of apoptotic cells in cell populations exposed to a continuous flow of different doses of TNF-α in the microfluidics chamber *without* Hoechst/UV imaging. In Figure 1—figure supplement 7 we show GFP-imaging fields from time-lapse acquisitions for 0.1, 1, and 10 ng/ml TNF-α continuous stimulation. The appearance of dead cells is considerably anticipated as the dose of TNF-α increases: the first dead cells appear on average at 12, 9 and 3 hours, respectively (red frames).

FACS analysis of the cell cycle supports this conclusion while excluding Hoechst toxicity at population level. Untreated cells or cells exposed to Hoechst, in the presence or absence of 10ng/ml of TNF, showed no global perturbation of the cell cycle but apoptotic cells appeared after three hours of incubation with TNF-α (Figure 1—figure supplement 10). Interestingly, a similar phenomenon was also observed by the Tay group: cells under static stimulation with 10 ng/ml TNF-α (movie S1 in (Tay et al., 2010) undergo almost no cell death, while many cells under constant TNF-α flow die (movie S1 in new Kellogg & Tay, 2015). This probably depends on culture conditions and on the cell type; the precise mechanism will be the object of future investigation.

Sustained versus damped oscillations:

The reviewers are concerned that in our culture conditions oscillations were “damped” and thus different from the “sustained” oscillations that were observed for the same cells in Sung et al. 2009 using different culture conditions. This is in part a semantic problem involving the terminology used. In Sung et al. 2009 oscillations of GFP-p65 were referred to as “sustained” as opposed to the extremely damped dynamics of NF-κB revealed by biochemical population-level assays as in the seminal paper of Hoffmann et al., 2002.

In the present work we do not claim that our cells have different dynamics from the one described in Sung et al. 2009 and Zambrano et al. 2014. We described the dynamics as “damped” as opposed to “sustained oscillations”, the latter referring to the regular series of peaks – almost sinusoidal – for time lapses longer than 12 hours reported by Kellogg and Tay, 2015.

In dynamical systems terms, damped oscillations are those that tend to stabilize to a fixed point while sustained oscillations converge to limit cycle. This is what we mean by damped oscillations here. In Figure 1—figure supplement 2, we show a representative collection of the heterogeneous dynamics observed under continuous flow of TNF-α, from non-oscillating cells to increasingly oscillating cells, as previously reported. This heterogeneity was summarized in the number of peaks per cell shown in Figure 1—figure supplement 3. This is remarkable provided that culture conditions are not static as in (Sung et al., 2009; Zambrano et al., 2014a), so autocrine-paracrine signaling, which might induce oscillations in late phases of the stimulation, is not possible.

Overall, though, the dynamics we observe cannot be described as sustained using Kellog and Tay terminology, provided that the peaks we find become more irregular and infrequent as the stimulation proceeds, in agreement with what observed previously. Most of our time series oscillate to apparently end up fluctuating around a value higher than the pre-stimulation level. In conclusion, we unfortunately used the same term “sustained” in two different meanings in two different papers, but there is little we can do: we should retroactively change the term used in Sung et al. 2009. We now try to define exactly in our manuscript the term sustained as we use it.

Possible interference by Hoechst and imaging on oscillations:

Finally, to further assess if labeling and imaging induce distortions of NF-κB dynamics, we decided to manually quantify NF-κB dynamics for cells that were not exposed to Hoechst nor imaged with UV. Examples of the manually derived single cell traces are shown in Figure 1—figure supplement 8, which show a heterogeneous but damped oscillatory dynamics as the ones shown in this manuscript and in our previous works, under imaging in the presence of Hoechst.

Actions taken:

In order to sharpen the manuscript, we have briefly delineated in the section “Periodic forcing turns heterogeneous NF-κB oscillations into synchronous oscillations” that controls were performed to exclude any interference between the signalling dynamics of NF-κB and the imaging conditions, and that cell death is mostly due to the continuous flow of TNF-α, referring to the movies. This is illustrated with several supplemental figures cited in the main text and thoroughly described in “Safety assessment for Hoechst staining” in the Materials and methods section.

To clarify further our description of the oscillations as damped, in section “Periodic forcing turns damped NF-κB oscillations into synchronous oscillations” we refer to the new figure, Figure 1—figure supplement 2, to illustrate the heterogeneous damped oscillatory dynamics that we observe as opposed to the sustained oscillations that are observed by Tay’s group. We also put our results in the context of previous observations done with the same cells.

*B) Importantly, the authors should compare the transcriptional response of cells with or without exposure to dye and UV light to verify that similar results are obtained under both conditions.*

This is actually an important point that needed clarification in the manuscript. In fact, we did perform our experiments of transcription with cells that were not exposed to Hoechst nor UV illuminated. As shown by the controls performed to address comment 1A, it is clear that the dynamics are not distorted by the imaging conditions, so neither should be the transcriptional response. However, in order to further exclude effects on transcription, we performed RT-PCR transcription assays for IκBα and Ccl5 in cells under constant TNF-α both with and without Hoechst staining. The results shown in Figure 5—figure supplement 2 indicate no significant difference between the two conditions.

Actions taken:

This is now clarified in the revised version of the manuscript. We also added in Materials and methods that no nuclear labelling was used in the transcription experiments, and that we performed controls showing that nuclear labelling wouldn’t have altered the results anyway. There we refer to the newly added Figure 5—figure supplement 2.

*2) With regards to the quantification of nuclear GFP-RelA, the authors state that the nuclear to cytoplasmic intensity, termed NCI, is an "internally normalized measure". Although this quantity has been used previously to represent nuclear localization of NF-κB (Nelson DE et al. Science 2004 and subsequent publications from the same group), more groups now approximate the nuclear concentration of NF-κB with the background-adjusted mean nuclear intensity (Lee TK* et al.

*2009 Science Signaling; Sung MH et al. 2014 Science Signaling; Lee REC, et al. 2014 Molecular Cell; Kellogg RA and Tay S, 2015 Cell), while minimizing the photobleaching which decreases fluorescent signals. NCI is not fully internally normalized, because both the numerator and the denominator fluctuate over time due to biological effects, not only due to technical effects. Would the same damped oscillations dynamics be observed with background-adjusted mean nuclear intensity?*

This comment made us realize that we did not provide convincing enough arguments of the validity of Nuclear to Cytoplasmic Intensity (NCI) as a quantifier of NF-κB dynamics and on the rationale of our selection. To summarize, we believe that the question raised by the reviewers can be answered with a clear yes and detail our reasoning below.

First, the imaging rationale: As we can see in some of the movies accompanying this manuscript (see e.g. Video 1 and Video 6) there are global fluctuations in the intensity of the images and of the background, which also vary in space. This is essentially due to small changes in the focus, slight inclination of the plates and changes in the laser intensity that might amplify-or reduce- the amount of light measured. These changes are sizeable when the signal to noise ratio is very low, as in the case of physiological levels of p65. Photobleaching, if any, is negligible: one can see in our movies – as in Video 1 – that cells are bright even in late phases of the imaging. Thus our rationale is to use a ratio of background-corrected intensities, so that these distortions will cancel out; this is the same rationale that we applied in our previous works, where we used ratios of intensities to quantify the dynamics (Sung et al., 2009; Zambrano et al., 2014a).

Second, the mathematical rationale: A precise description of our rationale for selecting NCI has been reported in the extended paragraph “Quantification of NF-κB dynamics” in the Materials and methods section. The mathematical discussion leads to conclude that, being a ratio of intensities, NCI is a faithful indicator of the ratio between nuclear and cytoplasmic NF-κB and hence robust to global changes in the image intensity. We use the same reasoning to show why the background-corrected average nuclear intensity would reflect the global imaging distortions in our settings and hence would not be an adequate readout of the nuclear amount of NF-κB. However, as the reviewers rightly point out, the numerator and denominator of this ratio can fluctuate due to biological reasons and this might blur or overestimate the oscillatory behavior of nuclear NF-κB, which is the truly important readout. If it were possible to assess that total NF-κB is constant, then the oscillations in the nuclear NF-κB would correspond to oscillations in NCI, as we demonstrate in Supplemental Information, due to the fact that the latter is a monotonously increasing function of the former, (Figure 1—figure supplement 8). We then provide evidence that NF-κB is indeed constant (see below).

Third, the biochemical rationale: Is the total amount of NF-κBa constant parameter? Although this has been widely assumed in the field and included in different mathematical models present in the literature, from the seminal paper (Hoffmann et al., 2002) to more recent papers as (Kellogg & Tay, 2015), this view has been challenged by a recent work (Sung et al., 2014) showing that p65 is regulated by a positive feedback in macrophages under LPS stimulation. Thus, to confirm that the “constant p65” assumption holds true for our GFP-p65 MEFs, we quantified p65 in cells stimulated with 10 ng/ml TNF-α for 8 hours with sampling every hour. Immunoblots in Figure 1—figure supplement 9 show no large changes in the p65 levels upon stimulation. We did not quantify p65 at single cell level using microscopy because fluctuations of our experimental setup might lead to imprecise results. However, we consider our assumption valid and in line with the literature.

Finally, variations in the areas of the nucleus and the cytoplasm might introduce small distortions, although they typically vary slowly. This problem can be easily solved by discarding cells for which the areas change abruptly – an indicator of imminent mitosis or cell death –, as our software does.

To further support the validity of NCI we are providing additional data.

First, we have that NCI(t) remains constant at single cell level – except for some spontaneous oscillations, as previously reported (Zambrano et al., 2014a) – as shown in Figure 1—figure supplement 8. This is remarkable because the movie from which these series were obtained shows considerable decrease in image intensity (see Video 6). Along the same lines we do not find upwards or downwards average trends in our data of NCI time series for different conditions, which indicates further that the time series are properly normalized and reflect well the dynamics. We have also plotted side-by-side the NCI values and average nuclear intensity obtained by manual segmentation in Figure 1—figure supplement 8. For this experiment the imaging conditions were stable so the same kind of qualitative behavior is obtained, as expected. A last indicator of the sensitivity of the quantifier comes from its ability to detect even small oscillatory peaks, see Figure 3 and Figure 3—figure supplement 1 as an example for 0.1 ng/ml TNF-α.

Actions taken:

We have tried to make all this clearer in the revised manuscript. In the first section of the results, where we briefly describe our approach, we explain that NCI is a valid quantifier of NF-κB dynamics while being robust to imaging distortions. Also, we decided to abandon the “internally normalized” term, which might be misleading.

Finally, in the Materials and methods section, we have introduced a new paragraph entitled “Selection of the quantifier of NF-κB dynamics” where we reproduced the discussion above on the validity of each quantifier providing the appropriate mathematical demonstration. The cited movie and supplement figures have been included in the new manuscript.

*3) Regarding the comparison with the recent report by Kellogg and Tay (2015 Cell), one important difference that is not highlighted here is that the applied pulse characteristics are different. The Tay group delivered a pulse by an infusion of TNF-*α

*without washing (i.e. T2 = 0, using the terminology of Figure 1), assuming a clearance process through internalization and degradation. Zambrano et al. enforced a washing and reset period between consecutive pulses. While it is a matter of debate which method better mimics a physiologically relevant situation, the different forcing protocols may have affected their results and make a direct comparison problematic. Could this explain why the two groups observe different results?*

We thank the reviewers for giving us the opportunity to expand on this point, which now has been addressed in the revised manuscript, also by adding new experimental results.

Indeed a crucial difference of the setup of Kellogg and Tay with respect to ours is that they periodically and quickly refresh the medium in contact with the cells, after which they assume that a clearance process takes place. This should produce a sawtooth TNF-α profile. We, instead, do constantly control the amount of TNF-α in direct contact in the cell by subjecting them to a *continuous* flow of the selected doses D_1_ and D_2_ for different times T_1_ and T_2_. Importantly, as we already pointed out, our approach also should reduce to the minimum the contribution of autocrine-paracrine signaling in the dynamics.

To gain further insights in whether these differences could explain the observed differences in the synchronization mechanism, we have measured the dynamics of our cells under sawtooth-like profiles of TNF-α with different periodicities. The results are shown in the new Figure 4—figure supplement 4. GFP-p65 MEFs synchronize to sawtooth pulses with T_f_=90'. In contrast to Kellogs and Tay’s results, our MEFs synchronize also to T_f_=60'. However, the synchronization after the first pulse is quickly lost for T_f_=180'.

Interestingly, with the sawtooth approach synchronizations are less sharp and reminiscent of those obtained using alternating doses (D_1_>0 and D_2_>0) for T_f_=60' (Figure 4—figure supplement 1), T_f_=90' (Figure 2—figure supplement 3) and T_f_=180' (Figure 3—figure supplement 2). These observations suggest that culturing conditions, absence of flow and chamber geometries possibly affect TNF-α decay and clearance. Moreover, autocrine-paracrine signaling may further contribute to distortions of the dynamics.

However, coming back to the point raised by the reviewers, the shown ability of our cells to synchronize to a sawtooth of T_f_=60', contrarily to what is reported by Kellogg and Tay (2015 Cell) for a similar forcing, further suggests that the synchronization mechanism strictly depends on cellular and environmental factors.

Actions taken:

We have included the new Figure 4—figure supplement 4 in the manuscript and discussed it in the section The synchronization mechanism of GFP-p65 knock-in MEFs is not entrainment. We have put this figure in the context of our discussion, stating that it provides further confirmation to our interpretation of the synchronization mechanism as that of a damped oscillator to a (sufficiently) strong external periodic forcing.

4) Concerning the choice of pulse duration for different forcing frequencies, the authors should justify why different "T1" periods were used for different forcing frequencies (45 min for 90 min pulses, 30 min for 180 min pulses, etc.). Michael White's group has reported that the temporal profile of nuclear NF-κB depends on the pulse duration (Ashall L et al. 2009 Science, FigureS7). Fixing the pulse duration might simplify the interpretation of results from a systematic comparison.

We agree with the reviewers: using the same value of in all the experiments would help in the interpretation of the results. A comprehensive analysis of many different combinations of T1, T2, D1 and D2 would have been extremely time-consuming and economically challenging. Therefore we started exploring only few combinations. We discuss here the reasons of our selection of T_1_ in each experiment while providing additional supplementary experiments to facilitate data comparison, and reinforcing the central message of the dynamics part of our work: that NF–κB oscillations can lock to a wide variety of input stimuli. This message has been highlighted further in the manuscript.

In our first experiments with T_f_ =90 min we decided to use T_1_ =45 min to see whether a "long square wave" forcing could give rise to synchronous dynamics similar to the T_1_ =5 min short pulses documented in the paper by Michael White's group (Ashall et al., 2009). To explore lower frequencies, like T_f_ =45 min, smaller values of T_1_have been chosen. We first tried with T_1_ =30 min, T_2_=15 min, (Figure 3, upper panels) and found that synchrony to the forcing was very poor, which highlighted the importance of a resetting phase in order to obtain synchronization to the input. By reducing the value of T_1_ =30 min to the value of T_1_ =22,5 min shown in Figure 3, we find the synchrony of the oscillations for different doses.

On the other hand, we selected T_1_=30 min for the experiment with T_f_ =180 min shown in Figure 3 because we observed that T_1_=30 min for T_f_ =90 min produced synchronous dynamics (Figure 3—figure supplement 3, lower panels), almost as clear as the ones observed for T_1_ =45 min. We also speculated that T_1_ =30 min,T_f_ =180 min might produce even sharper oscillations in the transcriptional output, which in turn was useful for Figure 5 and 6.

Actions taken:

We have better motivated our selection of T_1_in the manuscript and have included Figure 3—figure supplement 3 where the interested reader can find graphs on the dynamics for the conditions described above. Furthermore, we have highlighted the importance of the resetting time in obtaining the synchrony that we observe in the T_f_ =45 min, which is something that we overlooked in the original version of our manuscript.

*5) With regards to the computational model, one concern is that many parameter values seem ill-defined experimentally. Although the model builds on a previously published version, a more exhaustive presentation of the model (to explain which parameters are fixed, and which are fitted) should be included. It is also recommended that experimental effort be made to narrow down the number of free parameters: this greatly improves the quality of biochemical models, and makes for better predictions. Finally, IκBα is encoded in the model to exert a "transcriptional repressor" role although the cited study does not present particularly convincing evidence for any direct repressor activity of IκBα. Is this term necessary to recapitulate the experimental data, especially for a minimal mathematical model?*

Model description:

We agree with the reviewers that many parameters of the model are ill-defined: there is always a trade-off between the complexity of the model designed and the number of parameters that will be well defined, provided that any simplification implies to conflate in few processes many different ones. This is the case of our 8-dimensional mathematical model, but this holds true also for higher dimensional mathematical models of the NF-κB system, as e.g. Tay et al., 2010 and Ashall et al., 2009 – with twice as many variables as ours – where still ill-defined parameters can be found. As the reviewers implicitly suggest, a clear description of how the model is built can help to find a correspondence between variations in the experimental conditions and the system’s parameters. Hence, we have introduced in the Materials and methods a detailed description of the processes that our mathematical model takes into consideration, reproducing the basic discussion of the work where we described a previous version of the model (Zambrano et al. 2014b), including the normalizations performed, which leads to a simplification and a reduction of the free parameters.

Parameter variation:

Along the same lines, we have followed what we did in (Zambrano et al., 2014b) and assigned to each of the model parameters an uncertainty degree *D*, which is low *D=0.2* for parameters derived from biochemical rates of the literature (in our case, mostly from Tay et al., 2010, such as decay rates of proteins and RNA) and high (D=*1*) for others that are unknown or that correspond to processes that have been simplified, as in our case those involved in the A20 negative feedback. Parameters are then allowed to be varied by a random factor in the [10^-*D*^, 10*^D^*] interval, which means that the largest uncertainty degree imply a variation of up to one order of magnitude around the initial parameter value. In the new Table 2 we now explicitly state the biochemical rates and constants used, their origin, the parameters of the model to which they are related and the uncertainty degree considered for each of them. Importantly, in our fittings we allow each of the parameters to vary only by a factor in the [10^-*D*^, 10*^D^*] interval. This is now clearly stated in the Materials and methods.

Actions taken:

We also have clarified the notation of the parameters making explicit those that are fixed and those that are fitted. We essentially have three sets of parameters: P_S_, P_NF-_κ_B_, and P_G_. The first, P_S_, accounts for the parameters of the external time-varying signal; P_NF-_κ_B_, is the set of parameters that regulate NF-κB dynamics via the double IκBα–A20 negative feedback loop, and P_G_ encloses the parameters used to reproduce the gene expression. Thus, the mathematical description of the expression of two genes under the control of NF-κB under the same external forcing conditions would have the same values of P_S_ and P_NF-_κ_B_, but different –individually fitted – values of P_G_. In Figure 5, then, the same values of P_NF-_κ_B_ where used to reproduce the NCI, IκBα and CCL5 profiles, but for each of these a different P_G_ was used. Similarly, in the fittings of Figure 6 we use the same values of P_S_ and P_NF-_κ_B_as in Figure 5 but then each single gene is fitted with its own P_G_. This is also clearly explained in the new version of the manuscript, as well as in the captions of Figure 5 and 6.

On the role of IκBα as a repressor:

Finally, concerning the role of IκBα as a repressor, we agree that the evidence provided in (Arenzana-Seisdedos et al., 1995) is not conclusive. However, the role of IκBα as a repressor has been considered in a number of works, e.g. (Kellogg & Tay, 2015; Lipniacki et al., 2006; Tay et al., 2010), while not in others (Ashall et al., 2009; Sung et al., 2014). We have tested if our model can reproduce observed profiles even if such transcriptional repressor is not present, when simulating both the dynamics of NF-κB (Figure 6—figure supplement 7) and gene expression (Figure 6—figure supplement 7). Gene expression profiles are indeed reproduced, for genes with both increased and decreased transcription, with relative errors comparable to those obtained with our original model (Figure 6—figure supplement 7). It can also be observed that the degradation rate is the key parameter to reproduce the dynamics of each cluster of genes (Figure 6—figure supplement 7). Importantly, this exploration made us realize that for clusters 4-6 containing genes with decreased transcription we were essentially fitting the gene expression profiles as a simple RNA degradation process.

Actions taken:

All this has now been clarified in the text, where we also mention the newly added Figure 6—figure supplement 7. We also clarify that the role of IκBα as a repressor is not fundamental.

*6) With respect to the sensitivity of model behavior to parameter values, the authors acknowledge large parameter sensitivity of their model, with 2-fold changes in expression levels of proteins switching cells from damped to sustained oscillatory responses (c.f. Figure 1—figure supplement 5). A) An exact description of the parameters that were varied in these supplementary figures needs to be included.*

Using the notation that we have introduced to address point number 5, we have now specified that the parameters varied were P_NF-_κ_B_, according to the uncertainty degree that we assigned to each of them. We have added a similarly precise description of the parameters varied in the remaining numerical supplemental figures.

*B) More generally, a systematic parameter sensitivity of the model (e.g. in terms of generating varied types of oscillation) should be performed to allow readers to understand the robustness of the model (in terms of its predictive power). In particular, this could potentially identify key proteins whose up/down regulation would most affect the response phenotype. In turn, the natural variability in the expression levels of these signaling components should be taken into account: it would explain the phenotypic variability that must exist within the isogenic population of cells used in these experiments, but that was overlooked so far.*

Following the reviewers’ advice, we have performed a numerical exploration of our model in order to identify the variety of oscillatory behaviours that it can display. We varied the parameters of the model according to the uncertainty degree assigned to them. Since our model is a system of ordinary differential equations, it is possible to determine numerically the stability of the fixed points of the flow determined by equations (72)-(5) (see Materials and methods), and thus what dynamics should arise for each parameter combination.

We did generate a library of 10,000 random parameter combinations in the intervals prescribed, and did pre-select those that gave a response in the time series of *N(t)*. The eigenvalues of the fixed point are plotted in the complex plane in Figure 1—figure supplement 5; those in red correspond to sets of eigenvalues for which at least one of them has a real positive part. In that situation, the system converges to sustained oscillations, but the fraction of those combinations is less than 10% of the total, as shown in Figure 1—figure supplement 5. Examples of sustained oscillations similar to the ones shown by Kellogg and Tay 2015 and some spiky oscillation, similar to the ones shown in Ashall 2009, are shown in Figure 1—figure supplement 5.

We also find that most of the trajectories have four eigenvalues with imaginary parts, see Figure 1—figure supplement 5; this implies that most of the trajectories have two competing frequencies. For this reason, we find examples in which the interaction of fast and slow oscillating timescales can be observed. Additional examples of the variety of time series present can be shown in panel Figure 1—figure supplement 5. The fact that the signaling of NF-κB can present rich dynamics was pointed out in Sung et al. 2009. We hope that our results will contribute to further illustrate this important point that shows how even in isogenic populations heterogeneous dynamics – due to different parameter combinations – might arise.

Finally we present in Figure 1—figure supplement 5 the intervals of the parameters that give rise to sustained or damped oscillations (recall that the parameters selected are those giving trajectories responsive to the stimulus). We find that in our simple model the parameter ranges are very similar in oscillating and non-oscillating trajectories. Only for certain parameters related to the A20 feedback (n and dA) we find a clear difference. This suggests that variations in this upstream negative feedback can lead to strong variations in the type of dynamics. Overall, though, our results suggest that only the precise combination of parameters is what determines whether the system displays sustained or damped oscillations. Only further analytical work on the structure of bifurcations in parameter space may shed light on this important point. This type of work could also pave the way for targeted experiments that might be able to identify key proteins in NF-κB regulation, but this is far from the scope of the present work.

Actions taken:

We have tried to make all this clearer in the revised manuscript. An appropriate description has been added to the first section of the Results. The details of the stability analysis have been added to the Materials and methods section. Figure 1—figure supplement 5 has been improved with panels summarizing the discussion above.

C) Most relevant to the biology of NFκB signaling, such parameter sensitivity would help reconcile the diverse dynamics of NFκB response depending on the cell type under consideration (c.f. comparison with Kellogg and Tay, 2015 study). For that reason, it would be particularly useful to document systematically the variety of phenotypes (from damped oscillations, to spontaneous oscillations etc.) to predict and explain how different cell types use NFκB response differently to generate different functional responses, and to support the authors' argument that damped oscillations responses may be common while the stable oscillations may only be a rare case.

We believe that some the results previously shown and the revised version of the manuscript can address this point. In response to Point 2 we generated the new Figure 1—figure supplement 2, where we document the variety of phenotypes that we observe with examples of trajectories with different degrees of dampening, although the majority of them are those with few peaks (as shown in Figure 1—figure supplement 3). This indeed illustrates that in an isogenic population different oscillatory phenotypes can be observed. Hence, as the reviewers point out, it should be expected that also different cell types will display different oscillatory behaviors: this is probably the case of those studied in the work by Kellogg and Tay 2015, but the same could applied to recently published results with fibroblasts (see e.g. Figure 5, (Cheng et al., 2015)) or macrophages (see e.g. Figure 4 in Sung et al., 2014), that present also remarkable variability of behaviors. Furthermore, the numerical exploration performed to address the previous point illustrates how variations in the parameter combinations can give rise to widely different dynamics, which hints in the same direction. We can only claim that in our cells damped oscillations are much more common; our model suggests that damped oscillations are more frequent. Only the study of a wide variety of cell types can tell which of the two types of dynamics is more frequent. This will be hopefully the direction of our future work.

Actions taken:

All this has been made clearer in the first Results section and in the Discussion, where we state that our results (numerical and experimental) can also provide clues to reconcile the visions emerging from works where different cell types were considered, that we cite in the new version.

[Editors' note: further revisions were requested prior to acceptance, as described below.]

*[…] Only a few minor issues are noted for the revised version which we hope can be rapidly addressed by the authors. Cell death, cell division and imaging conditions* – *overall, the additional data provided by the authors, particularly Figure 1—figure supplement 7 and Figure 5—figure supplement 2 do suggest that the cell death that is observed in some of the experiments is likely due to the presence of TNF (under flow condition and in the absence of serum), and not just an artefact of the use of a Hoescht dye and UV illumination to identify the nucleus of each cell. However, the evidence from PO4-gammaH2AX in Figure 1—figure supplement 6 has an important weakness and should probably be de-emphasized. Indeed, this epitope marks double strand breaks (DSBs) and while doxorubicin (positive control) is known to produce DNA double strand breaks (DSB) in cells in most phases of the cell cycle (DSB are linked to genes actively transcribed; e.g. http://www.ncbi.nlm.nih.gov/pubmed/25705119) UV illumination creates other types of DNA damage (pyrimidine dimers) but only lead to DSBs in cells in S-phase and on a longer time scale than that used bythe authors in this figure (6-12 hrs, vs. 3 hrs* – *e.g. http://www.ncbi.nlm.nih.gov/pubmed/20947453 and http://www.ncbi.nlm.nih.gov/pubmed/18800352). Therefore, lack of PO4-gammaH2AX staining does not necessarily preclude the possibility of UV-induced DNA damage.*

To address this, we assessed the presence of thymine dimers as a specific indicator of DNA damage upon UV exposure. Thymine dimers in imaged cells were at a level below the sensitivity threshold of the immunostaining method. Now results are reported in a new figure (Figure 1—figure supplement 6).

We agree with the reviewers’ comment on the role of gammaH2AX. The text in the Materials and methods has been changed to de-emphasise the relevance of this marker for photo-damage. Appropriate new references have been added.

*Square vs. sawtooth stimulation profile. Because the geometry of the microfluidic chamber is very different in this study from that of the device used in Kellogg and Tay, it is unclear if a significant reduction in TNF concentration is achieved when medium flow is stopped (as noted by the authors, TNF is stable in their chamber, and would decay only if actively degraded by cells). Importantly, this could be an alternative explanation for the lack of synchrony* – *this stimulation profile may be closer to static concentration than to oscillatory stimulation* –

*which should be noted in the discussion of these data.*

This is an insightful comment. Indeed, the different geometry of the chamber and presumably different culture conditions can lead to a different decay rates for the TNF-α when put in contact with the cells. This observation has been included in the discussion of the sawtooth profile data.

*Regarding the sensitivity analysis of the model* –

*the sensitivity analysis does provide a greater context for the observations by the authors as well as the cited work by Kellogg and Tay. The authors mention that parameters related to the A20 feedback are the only one for which there is a clear difference between values that lead to different types of dynamics. When discussing this result, the authors may find it helpful to refer to Werner et al. (2008; Genes and Dev.), where the A20 feedback was explored computationally and experimentally. This paper should also be cited in the fourth paragraph of the Introduction, when introducing the A20 feedback.*

We thank the reviewers for this comment. Indeed, the paper mentioned is one of the first descriptions of how the interplay between the A20 and IκB regulatory modules provides a fine-tuning of NF-κB response to an external stimulus. We now cite it when briefly describing the A20 feedback in the introduction. We also cite it when discussing the role of A20 in the type of dynamics observed in the paragraph “Classification of the different dynamical responses for constant stimulus”of the“Mathematical modelling” section of the Materials and methods. Finally, we also cite it in the Discussion when commenting on the same results.

*Regarding the role of IκBα as a repressor* –

*the new results from the authors do shed more light on that issue. However, the authors should review their model diagram as some are ambiguous or possibly misleading. It would be easier for the readers to interpret the model if the "activation" arrow for NF-κB and the "inhibition" arrow for IκBα were carefully positioned to target the reactions on which they act.*

The diagrams in Figure 5, Figure 1—figure supplement 4 and Figure 6—figure supplement 7 have been changed as suggested. Furthermore, the figure captions of these figures now state clearly the meaning of the green and red arrows (transcriptional activation and repression, respectively).

In the third paragraph of the subsection “Periodic forcing turns heterogeneous NF-κB oscillations into synchronous oscillations”:

*a) "Replicate and show…"; "divide" or "proliferate" would be a better term to characterize the cells' behavior.*

We have changed the text accordingly.

*b) “Limited cell death… "; it may be more appropriate to just state "cell death…".*

The text has been changed.

*In the fourth paragraph of the Introduction: "In resting cells, p65:p50 exists mostly as a cytoplasmic complex bound to the IκB inhibitors (Hoffmann et al., 2002)" would be better rephrased to: "In resting cells, p65 exists mostly within a cytoplasmic complex bound to the IκB inhibitors (Hoffmann et al., 2002)."*

This has been rephrased as suggested.

*"Figure 4*: *The authors should indicate in the main text or in the figure legend how many forcing cycles the cells were stimulated with before withdrawal of the forcing." The authors have now added cycle numbers on the figures although it would be helpful if the figure legend explained what these numbers represent.*

Now we describe in the legend of Figure 4 what those numbers represent.